# Corticofugal regulation of predictive coding

**Alexandria MH Lesicko[1], Christopher F Angeloni[2], Jennifer M Blackwell[1,3], Mariella De Biasi[4,5,6], Maria N Geffen[1,6,7]\***

[1]Department of Otorhinolaryngology, University of Pennsylvania, Philadelphia, United States; [2]Department of Psychology, University of Pennsylvania, Philadelphia, United States; [3]Department of Neurobiology and Behavior, Stony Brook University, Stony Brook, United States; [4]Department of Psychiatry, University of Pennsylvania, Philadelphia, United States; [5]Department of Systems Pharmacology and Experimental Therapeutics, University of Pennsylvania, Philadelphia, United States; [6]Department of Neuroscience, University of Pennsylvania, Philadelphia, United States; [7]Department of Neurology, University of Pennsylvania, Philadelphia, United States

**\*For correspondence:**
mgeffen@pennmedicine.upenn.edu

**Competing interest:** The authors declare that no competing interests exist.

**Abstract** Sensory systems must account for both contextual factors and prior experience to adaptively engage with the dynamic external environment. In the central auditory system, neurons modulate their responses to sounds based on statistical context. These response modulations can be understood through a hierarchical predictive coding lens: responses to repeated stimuli are progressively decreased, in a process known as repetition suppression, whereas unexpected stimuli produce a prediction error signal. Prediction error incrementally increases along the auditory hierarchy from the inferior colliculus (IC) to the auditory cortex (AC), suggesting that these regions may engage in hierarchical predictive coding. A potential substrate for top-down predictive cues is the massive set of descending projections from the AC to subcortical structures, although the role of this system in predictive processing has never been directly assessed. We tested the effect of optogenetic inactivation of the auditory cortico-collicular feedback in awake mice on responses of IC neurons to stimuli designed to test prediction error and repetition suppression. Inactivation of the cortico-collicular pathway led to a decrease in prediction error in IC. Repetition suppression was unaffected by cortico-collicular inactivation, suggesting that this metric may reflect fatigue of bottom-up sensory inputs rather than predictive processing. We also discovered populations of IC units that exhibit repetition enhancement, a sequential increase in firing with stimulus repetition. Cortico-collicular inactivation led to a decrease in repetition enhancement in the central nucleus of IC, suggesting that it is a top-down phenomenon. Negative prediction error, a stronger response to a tone in a predictable rather than unpredictable sequence, was suppressed in shell IC units during cortico-collicular inactivation. These changes in predictive coding metrics arose from bidirectional modulations in the response to the standard and deviant contexts, such that the units in IC responded more similarly to each context in the absence of cortical input. We also investigated how these metrics compare between the anesthetized and awake states by recording from the same units under both conditions. We found that metrics of predictive coding and deviance detection differ depending on the anesthetic state of the animal, with negative prediction error emerging in the central IC and repetition enhancement and prediction error being more prevalent in the absence of anesthesia. Overall, our results demonstrate that the AC provides cues about the statistical context of sound to subcortical brain regions via direct feedback, regulating processing of both prediction and repetition.

## Editor's evaluation

This study concerns the neural representation of prediction in the central auditory pathway. The authors report that top-down inputs from the auditory cortex carry contextual cues that enable subcortical neurons to distinguish between predictable and unexpected sounds. This work provides important insights into how feedback pathways in the auditory system modulate feedforward signals in a context-dependent fashion.

## Introduction

Sensory systems differentially encode environmental stimuli depending on the context in which they are encountered (*De Franceschi and Barkat, 2020*; *Herrmann et al., 2015*; *Jaramillo et al., 2014*; *Pakan et al., 2016*; *Takesian et al., 2018*; *Zhai et al., 2020*). The same physical stimulus can elicit distinct neuronal responses depending on whether it is predictable or unexpected in a given sensory stream (*Weissbart et al., 2020*; *Yaron et al., 2012*). Neurons in select regions of the central auditory system are sensitive to statistical context, responding more strongly to a tone when it is presented rarely (a 'deviant') than when it is commonplace (a 'standard') (*Ulanovsky et al., 2003*). This phenomenon, known as stimulus-specific adaptation (SSA), is prevalent in the auditory cortex (AC) (*Natan et al., 2015*; *Ulanovsky et al., 2003*). Weaker SSA is present in regions peripheral to the AC, including the auditory midbrain, or inferior colliculus (IC), and the auditory thalamus, or medial geniculate body (MGB) (*Anderson et al., 2009*; *Antunes et al., 2010*; *Duque and Malmierca, 2015*; *Malmierca et al., 2009*; *Taaseh et al., 2011*; *Ulanovsky et al., 2003*). Subdivisions in IC and MGB that receive descending projections from AC exhibit relatively higher SSA levels than their lemniscal counterparts (*Antunes et al., 2010*; *Duque et al., 2012*), suggesting that SSA may be generated de novo in AC and subsequently broadcast to subcortical structures via corticofugal projections (*Nelken and Ulanovsky, 2007*). Silencing of AC through cooling, however, has been shown to modulate, but not abolish, SSA in IC and MGB of anesthetized rats (*Anderson and Malmierca, 2013*; *Antunes and Malmierca, 2011*).

Recent studies have implemented additional control tone sequences to further decompose the traditional SSA index into two distinct underlying processes: repetition suppression and prediction error (*Harms et al., 2014*; *Parras et al., 2017*; *Ruhnau et al., 2012*). Repetition suppression is characterized by a decrease in firing rate to each subsequent presentation of a standard tone, whereas prediction error signals an enhanced response to a deviant tone (*Auksztulewicz and Friston, 2016*; *Parras et al., 2017*). Hierarchical predictive coding posits that prediction errors signal the mismatch between predictions, formed based on prior experience with repeated presentations of the standard, and actual sensory input in the presence of a deviant (*Friston, 2009*; *Friston and Kiebel, 2009*). These predictions are generated at higher levels of the sensory hierarchy and broadcast to lower stations to minimize processing of redundant input and maximize coding efficiency (*Friston, 2009*; *Friston and Kiebel, 2009*). Prediction error has been proposed to underlie true deviance detection, while repetition suppression is thought to potentially reflect synaptic depression (*Parras et al., 2017*; *Taaseh et al., 2011*). Prediction error increases along the auditory hierarchy and is more prevalent in regions of IC and MGB that receive cortical feedback (*Parras et al., 2017*), suggesting that these subcortical regions may engage in hierarchical predictive coding, with AC potentially providing predictive cues to IC and MGB. However, how feedback projections from AC shape predictive processing in subcortical targets has never been directly assessed. In fact, virtually all models of hierarchical predictive coding to date have focused on intracortical connections, with the massive system of descending corticofugal projections remaining unexplored (*Asilador and Llano, 2020*; *Bastos et al., 2012*).

Here, we investigated how inputs from AC to IC, the first station in the auditory system in which prediction error is found, shape metrics associated with predictive coding and deviance detection (*Parras et al., 2017*). To test this, we optogenetically inactivated cortico-collicular feedback while recording neuronal responses in IC and found that prediction error, negative prediction error, and repetition enhancement in IC are altered in the absence of cortical input. Our results suggest that the cortico-collicular pathway sends cues from AC to IC regarding the statistical context of auditory stimuli.

## Results

### Experimental design

We used a Cre/FLEX viral injection strategy to selectively express the inhibitory opsin, ArchT, in cortico-collicular neurons of four mice by injecting a retroAAV-Cre-GFP construct into IC and an AAV9-FLEX-ArchT-tdTomato construct into AC (*Figure 1A*, left). The retroAAV-Cre-GFP construct is transported in a retrograde fashion and expressed in neurons that project to IC (*Blackwell et al., 2020*). The genes encoded in the AAV9-FLEX-ArchT-tdTomato construct can only be expressed in neurons containing the Cre construct, thereby limiting ArchT expression to neurons in AC that project to IC. In the presence of green light, ArchT, a light-driven outward proton pump, mediates rapid, reversible inactivation of the neurons in which it is expressed (*Han et al., 2011*).

We implanted cannulas over AC in mice injected with the Cre/FLEX constructs and a 532 nm laser was used to provide green light illumination to the region, allowing for inactivation of cortico-collicular neurons (*Figure 1A*, right). The mice were head-fixed and a 32-channel probe was lowered into IC to perform awake extracellular recordings (*Figure 1A*). Auditory stimuli consisted of oddball sequences of two repeated pure tones, presented at a 90:10 standard-to-deviant ratio and half-octave frequency separation (*Figure 1B*). On a subset of trials, presentations of either the deviant or the last standard prior to the deviant were coupled with activation of the green laser (*Figure 1B*, right).

Units that displayed a significantly higher response to the deviant than the standard were designated as 'adapting' units, while those that exhibited a significantly higher response to the standard than the deviant were categorized as 'facilitating' units (*Figure 1D*). The difference in firing rate to the standard and deviant was quantified with an index of neuronal mismatch (iMM), which is equivalent to the SSA index used in previous studies (*Parras et al., 2017*).

A cascade stimulus consisting of 10 evenly spaced tones, including the tone pair from the oddball sequence, was presented to further decompose the neuronal mismatch between the responses to the standard and deviant (*Figure 1C and D*). This stimulus is unique in that each tone occurs with the same likelihood as the deviant tone in the oddball stimulus (10%), but it contains no true statistical deviants: each tone has the same likelihood of presentation, and the tone sequence overall follows a regular and predictable pattern (*Parras et al., 2017*). Therefore, the response to a given tone when it is embedded in the cascade can be compared to the response when it is a deviant in order to isolate prediction error effects (*Figure 1C and D*, top). A neuron exhibits prediction error if it fires more strongly to a tone when it is a deviant than when it is presented in the cascade sequence (*Figure 1D*, top). Conversely, if a neuron responds more strongly to a tone presented in the cascade sequence than when it is a deviant, the neuron encodes negative prediction error (*Figure 1D*, bottom). This phenomenon is quantified using an index of prediction error (iPE), with positive indices indicating prediction error and negative indices representing negative prediction error (*Figure 1D*).

The cascade sequence is also free from repetition effects since adjacent tone presentations never include a tone of the same frequency (*Figure 1C*). Therefore, the response to a given tone embedded in the cascade sequence can be compared to the response generated when that tone is a standard. The difference in response indicates either repetition suppression (stronger response to the tone in the cascade) (*Figure 1D*, top) or repetition enhancement (stronger response to the tone as a standard) (*Figure 1D*, bottom). These contrasting processes are quantified by the index of repetition suppression (iRS), with a positive index indicating repetition suppression and a negative index representing repetition enhancement (*Figure 1D*).

### Cre/FLEX viral injection strategy enables selective inactivation of cortico-collicular neurons

Examination of fixed tissue from injected mice revealed that expression of the retroAAV-Cre-GFP construct was restricted to IC (*Figure 1—figure supplement 1A*, top left). Somatic expression of GFP (indicating the presence of Cre) was restricted to layer 5 and deep layer 6 of AC, which contain cortico-collicular cell bodies, and was broadly distributed throughout the rostro-caudal extent of AC (*Figure 1—figure supplement 1A*, right) (*Bajo et al., 2007*; *Schofield, 2009*; *Yudintsev et al., 2019*). Expression of tdTomato was found in the soma and processes of neurons in layers 5 and 6, with additional apical dendritic labeling observed in the upper cortical layers (*Figure 1—figure supplement 1A*, right). The laminar expression of tdTomato is consistent with previous studies and suggests that AAV9-FLEX-ArchT-tdTomato expression is Cre-dependent and not due to nonspecific

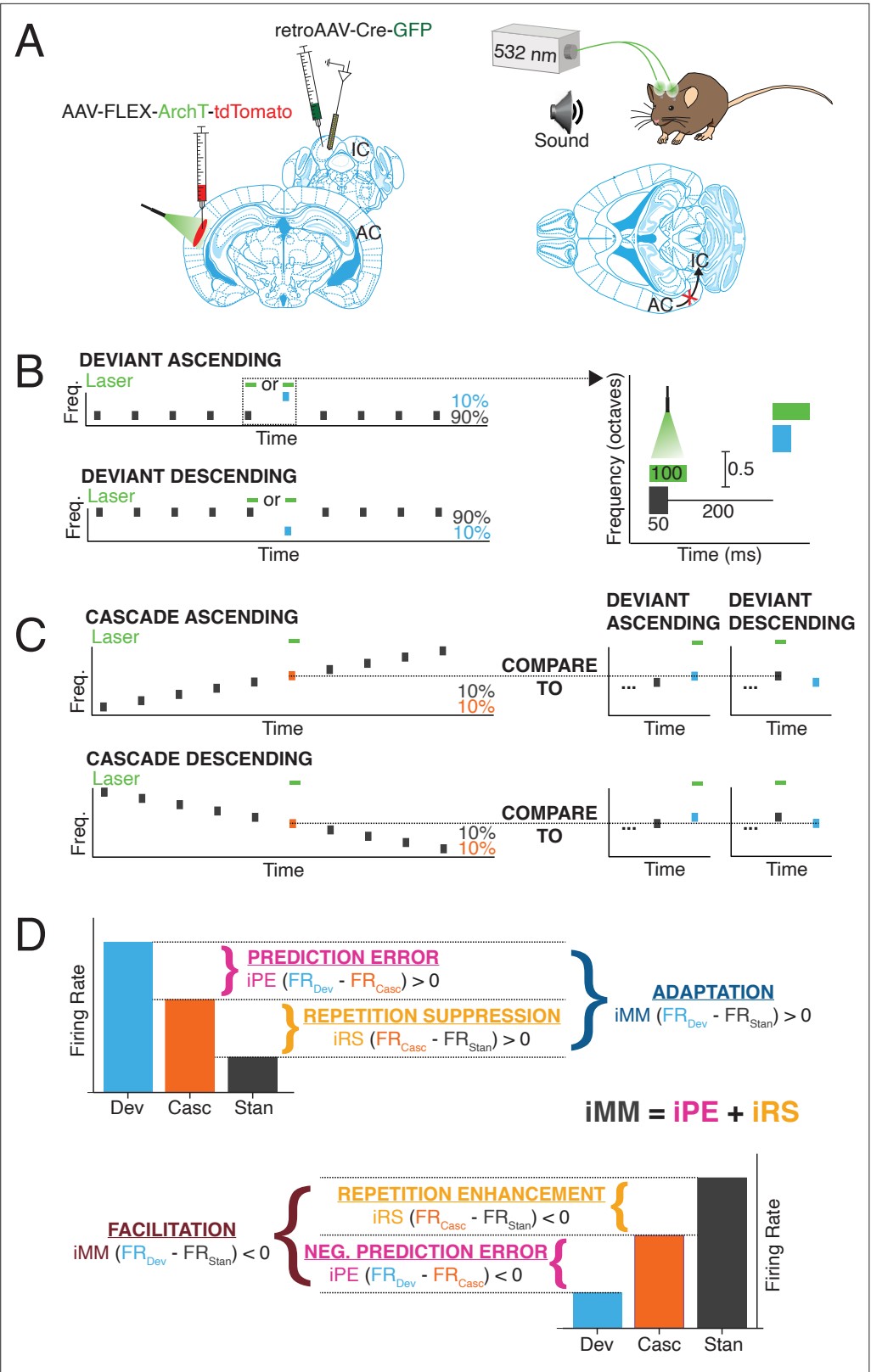

**Figure 1.** Experimental design. (**A**) Cre/FLEX dual injections for selective ArchT expression in cortico-collicular neurons. Recordings were performed in the inferior colliculus (IC) while inactivation was mediated by a 532 nm laser connected to cannulas implanted over the auditory cortex (AC). (**B**) Oddball stimuli consisted of pairs of pure tones separated by 0.5 octave with a 90:10 standard-to-deviant ratio. Two sequences were constructed such

*Figure 1 continued*

that each frequency is represented as both the standard and the deviant. (**C**) Cascade sequences consisted of 10 evenly spaced tones separated by 0.5 octaves, with both frequencies from the oddball sequence included in the sequence. Responses to tones in the cascade context were compared to responses in the standard and deviant context to analyze repetition and prediction effects, respectively. (**D**) A positive index of neuronal mismatch (iMM) (top diagram) indicates a stronger response to the deviant than the standard (adaptation), while a negative iMM (bottom diagram) indicates a stronger response to the standard than to the deviant (facilitation). The iMM can be further decomposed into an index of prediction error (iPE) and an index of repetition suppression (iRS). Positive iPE values represent prediction error, and negative values convey negative prediction error. Positive iRS indices indicate repetition suppression, while repetition enhancement is represented by negative values.

The online version of this article includes the following figure supplement(s) for figure 1:

**Figure supplement 1.** Cre/FLEX viral injection strategy enables selective inactivation of cortico-collicular neurons.

**Figure supplement 2.** Parsing of recording sites into central and shell locations.

---

labeling (*Blackwell et al., 2020*). Axons and terminals labeled with tdTomato were distributed in IC in a manner matching the known projection pattern of this pathway, with dense, 'patchy' labeling in shell regions of IC (*Figure 1—figure supplement 1A*, bottom left) (*Herbert et al., 1991*; *Lesicko et al., 2016*; *Saldaña et al., 1996*; *Torii et al., 2013*). These data confirm that our viral injection strategy leads to selective transfection of cortico-collicular neurons.

Extracellular recordings in AC of injected mice revealed a reduction in firing rate during the duration of the laser stimulus in several units (*Figure 1—figure supplements 1B and 2C*). In these putative cortico-collicular units, laser-induced inactivation led to a mean ~60% reduction in firing rate at baseline (*Figure 1—figure supplement 1C*, left; *Figure 1—figure supplement 2D*, top; *Table 1*; p=1.9e-06, Wilcoxon signed-rank test) and an average ~45% reduction in firing during presentation of pure tone stimuli (*Figure 1—figure supplement 1C*, right; *Figure 1—figure supplement 2D*, bottom; *Table 1*; p=1.9e-06, Wilcoxon signed-rank test). These results indicate that our optogenetic parameters significantly suppress cortico-collicular units.

## Parsing of recording sites into central and shell locations

Shell and central regions of IC differ in their tuning, degree of adaptation, and amount of input from AC, and may also play distinct roles in predictive processing (*Aitkin et al., 1975*; *Bajo et al., 2007*; *Blackwell et al., 2020*; *Duque et al., 2012*; *Herbert et al., 1991*; *Stebbings et al., 2014*; *Syka et al., 2000*). We quantitatively parsed our recording sites by exploiting known differences in the sharpness of tuning and direction of frequency gradients between shell and central regions: shell IC neurons tend to have broader frequency tuning (low sparseness) than central IC neurons, and the central IC is characterized by a highly stereotyped tonotopic gradient with depth (*Figure 1—figure supplement 2A*; *Aitkin et al., 1975*; *Chen et al., 2012*; *Malmierca et al., 2008*; *Stiebler and Ehret, 1985*; *Syka et al., 2000*). Similar to previously established procedures used in human and monkey IC research, we performed clustering analysis using the mean sparsity and variation in best frequency with depth from each recording site to determine whether it was from the central nucleus or shell regions of IC (*Figure 1—figure supplement 2B and C*; *Bulkin and Groh, 2011*; *Ress and Chandrasekaran, 2013*). In a subset of recordings, we also marked the recording electrode with a lipophilic dye to histologically confirm the recording location (*Figure 1—figure supplement 2D*).

IC units in both regions exhibited multiple response types to pure tone stimuli (*Figure 1—figure supplement 2E*). In addition to excitatory responses (e.g., onset and sustained responses), inhibited and offset responses were common, as has previously been characterized in IC of awake animals (*Figure 1—figure supplement 2E*, top right, bottom middle; *Duque and Malmierca, 2015*). Consistent with previous findings, tuning curves from central regions were sharp and narrow, whereas units in shell regions exhibited broad frequency tuning (*Figure 1—figure supplement 2F*, left vs. right; *Aitkin et al., 1975*; *Syka et al., 2000*). Inhibited side bands were common in tuning curves from both regions, and some inhibited tuning curves were observed (*Figure 1—figure supplement 2G*). These data confirm that our experimental parameters elicit sound responses and tuning properties characteristic of central and shell regions of the awake IC (*Aitkin et al., 1975*; *Duque and Malmierca, 2015*; *Syka et al., 2000*).

**Table 1.** Statistical comparisons for experimental data.

| Comparison | Figure | Mean | Median | SD | SEM | CI (±) | Test | Test statistic | N | df | p | Effect size |
|---|---|---|---|---|---|---|---|---|---|---|---|---|
| Response of putative cortico-collicular units in silence (laser OFF vs. ON) | **Figure 1—figure supplement 1D** (top) | OFF: 11 ON: 4.1 | OFF: 9.0 ON: 3.5 | OFF: 8.9 ON: 3.5 | OFF: 2.0 ON: 0.78 | OFF: 4.2 ON: 1.6 | Wilcoxon signed-rank test | V = 0 | 20 | NA | 1.9e-06 | 0.88 |
| Response of putative cortico-collicular units to pure tones (laser OFF vs. ON) | **Figure 1—figure supplement 1D** (bottom) | OFF: 18 ON: 9.6 | OFF: 8.8 ON: 4.3 | OFF: 24 ON: 12 | OFF: 5.4 ON: 2.7 | OFF: 11 ON: 5.6 | Wilcoxon signed-rank test | V = 0 | 20 | NA | 1.9e-06 | 0.88 |
| iMM central (awake vs. anesthetized) | **Figure 2B** | Aw: 0.050 An: 0.25 | Aw: 0.045 An: 0.28 | Aw: 0.21 An: 0.49 | Aw: 0.024 An: 0.074 | Aw: 0.047 An: 0.15 | Wilcoxon rank-sum test | W = 952.5 | Aw: 78 An: 43 | NA | 8.8e-05 | 0.36 |
| iPE central (awake vs. anesthetized) | **Figure 2C** | Aw: −0.13 An: 0.077 | Aw: −0.11 An: 0.098 | Aw: 0.17 An: 0.53 | Aw: 0.019 An: 0.081 | Aw: 0.038 An: 0.16 | Student's t-test | t = −2.5 | Aw: 78 An: 43 | 38 | 0.017 | 0.52 |
| iRS central (awake vs. anesthetized) | **Figure 2D** | Aw: 0.18 An: 0.18 | Aw: 0.17 An: 0.30 | Aw: 0.17 An: 0.56 | Aw: 0.019 An: 0.085 | Aw: 0.039 An: 0.17 | Wilcoxon rank-sum test | W = 1444 | Aw: 78 An: 43 | NA | 0.21 | 0.12 |
| iMM shell (awake vs. anesthetized) | **Figure 2E** | Aw: 0.095 An: 0.27 | Aw: 0.090 An: 0.27 | Aw: 0.31 An: 0.35 | Aw: 0.025 An: 0.022 | Aw: 0.050 An: 0.043 | Wilcoxon rank-sum test | W = 12,502 | Aw: 147 An: 254 | NA | 3.5e-08 | 0.28 |
| iPE shell (awake vs. anesthetized) | **Figure 2F** | Aw: 0.15 An: 0.018 | Aw: 0.15 An: −0.0075 | Aw: 0.33 An: 0.39 | Aw: 0.027 An: 0.025 | Aw: 0.053 An: 0.049 | Wilcoxon rank-sum test | W = 23,368 | Aw: 147 An: 254 | NA | 2.6e-05 | 0.21 |
| iRS shell (awake vs. anesthetized) | **Figure 2G** | Aw: −0.056 An: 0.25 | Aw: −0.085 An: 0.29 | Aw: 0.36 An: 0.33 | Aw: 0.029 An: 0.020 | Aw: 0.058 An: 0.040 | Wilcoxon rank-sum test | W = 9501.5 | Aw: 147 An: 254 | NA | 2.5e-16 | 0.41 |
| iMM central adapting (laser OFF vs. ON) | **Figure 3D** (top) | OFF: 0.26 ON: 0.21 | OFF: 0.24 ON: 0.19 | OFF: 0.096 ON: 0.13 | OFF: 0.013 ON: 0.019 | OFF: 0.027 ON: 0.037 | Wilcoxon signed-rank test | V = 1083 | 52 | NA | 0.00034 | 0.50 |
| iPE central adapting (laser OFF vs. ON) | **Figure 3D** (middle) | OFF: 0.0077 ON: −0.029 | OFF: 0.036 ON: 0.0041 | OFF: 0.16 ON: 0.16 | OFF: 0.022 ON: 0.022 | OFF: 0.043 ON: 0.044 | Wilcoxon signed-rank test | V = 907 | 52 | NA | 0.048 | 0.28 |
| iRS central adapting (laser OFF vs. ON) | **Figure 3D** (bottom) | OFF: 0.25 ON: 0.24 | OFF: 0.24 ON: 0.24 | OFF: 0.16 ON: 0.16 | OFF: 0.023 ON: 0.022 | OFF: 0.046 ON: 0.045 | Wilcoxon signed-rank test | V = 832 | 52 | NA | 0.19 | 0.18 |
| iMM shell adapting (laser OFF vs. ON) | **Figure 3E** (top) | OFF: 0.34 ON: 0.31 | OFF: 0.32 ON: 0.28 | OFF: 0.19 ON: 0.20 | OFF: 0.017 ON: 0.019 | OFF: 0.035 ON: 0.037 | Wilcoxon signed-rank test | V = 4283 | 113 | NA | 0.0023 | 0.29 |
| iPE shell adapting (laser OFF vs. ON) | **Figure 3E** (middle) | OFF: 0.15 ON: 0.14 | OFF: 0.12 ON: 0.10 | OFF: 0.30 ON: 0.30 | OFF: 0.028 ON: 0.028 | OFF: 0.056 ON: 0.056 | Wilcoxon signed-rank test | V = 3963 | 113 | NA | 0.034 | 0.20 |
| iRS shell adapting (laser OFF vs. ON) | **Figure 3E** (bottom) | OFF: 0.19 ON: 0.17 | OFF: 0.19 ON: 0.16 | OFF: 0.24 ON: 0.24 | OFF: 0.023 ON: 0.023 | OFF: 0.045 ON: 0.045 | Paired t-test | t = 1.6 | 113 | 112 | 0.11 | 0.15 |
| iMM central facilitating (laser OFF vs. ON) | **Figure 3G** (top) | OFF: −0.32 ON: −0.13 | OFF: −0.31 ON: −0.11 | OFF: 0.16 ON: 0.19 | OFF: 0.042 ON: 0.050 | OFF: 0.090 ON: 0.11 | Paired t-test | t = −3.5 | 14 | 13 | 0.0036 | 0.95 |
| iPE central facilitating (laser OFF vs. ON) | **Figure 3G** (middle) | OFF: −0.20 ON: −0.17 | OFF: −0.24 ON: −0.20 | OFF: 0.20 ON: 0.17 | OFF: 0.054 ON: 0.044 | OFF: 0.12 ON: 0.095 | Paired t-test | t = −1.2 | 14 | 13 | 0.25 | 0.32 |
| iRS central facilitating (laser OFF vs. ON) | **Figure 3G** (bottom) | OFF: −0.12 ON: 0.036 | OFF: −0.092 ON: 0.069 | OFF: 0.18 ON: 0.24 | OFF: 0.049 ON: 0.064 | OFF: 0.11 ON: 0.14 | Paired t-test | t = −3.7 | 14 | 13 | 0.0026 | 1.0 |
| iMM shell facilitating (laser OFF vs. ON) | **Figure 3H** (top) | OFF: −0.29 ON: −0.19 | OFF: −0.24 ON: −0.15 | OFF: 0.15 ON: 0.16 | OFF: 0.024 ON: 0.026 | OFF: 0.048 ON: 0.052 | Wilcoxon signed-rank test | V = 159 | 38 | NA | 0.0016 | 0.50 |
| iPE shell facilitating (laser OFF vs. ON) | **Figure 3H** (middle) | OFF: −0.026 ON: 0.033 | OFF: 0.011 ON: 0.023 | OFF: 0.26 ON: 0.29 | OFF: 0.042 ON: 0.047 | OFF: 0.085 ON: 0.096 | Wilcoxon signed-rank test | V = 227 | 38 | NA | 0.037 | 0.34 |
| iRS shell facilitating (laser OFF vs. ON) | **Figure 3H** (bottom) | OFF: −0.26 ON: −0.23 | OFF: −0.29 ON: −0.23 | OFF: 0.32 ON: 0.33 | OFF: 0.052 ON: 0.054 | OFF: 0.11 ON: 0.11 | Wilcoxon signed-rank test | V = 254 | 38 | NA | 0.093 | 0.27 |
| iMM central nonadapting (laser OFF vs. ON) | **Figure 4C** (top) | OFF: 0.022 ON: 0.072 | OFF: 0.023 ON: 0.065 | OFF: 0.12 ON: 0.14 | OFF: 0.0094 ON: 0.011 | OFF: 0.019 ON: 0.022 | Wilcoxon signed-rank test | V = 3419 | 155 | NA | 2.7e-06 | 0.38 |
| iPE central nonadapting (laser OFF vs. ON) | **Figure 4C** (middle top) | OFF: −0.096 ON: −0.081 | OFF: −0.098 ON: −0.093 | OFF: 0.19 ON: 0.19 | OFF: 0.015 ON: 0.015 | OFF: 0.030 ON: 0.030 | Wilcoxon signed-rank test | V = 5327 | 155 | NA | 0.20 | 0.10 |

*Table 1 continued on next page*

# Table 1 continued

| Comparison | Figure | Mean | Median | SD | SEM | CI (±) | Test | Test statistic | N | df | p | Effect size |
|---|---|---|---|---|---|---|---|---|---|---|---|---|
| iRS central nonadapting (laser OFF vs. ON) | **Figure 4C** (middle bottom) | OFF: 0.12 ON: 0.15 | OFF: 0.12 ON: 0.15 | OFF: 0.15 ON: 0.17 | OFF: 0.012 ON: 0.013 | OFF: 0.024 ON: 0.027 | Wilcoxon signed-rank test | V = 4224 | 155 | NA | 0.0011 | 0.26 |
| iRS > 0 central nonadapting (laser OFF vs. ON) | **Figure 4C** (bottom) | OFF: 0.17 ON: 0.19 | OFF: 0.16 ON: 0.18 | OFF: 0.10 ON: 0.15 | OFF: 9.1e-03 ON: 0.013 | OFF: 1.8e-02 ON: 0.026 | Wilcoxon signed-rank test | V = 3313 | 127 | NA | 0.071 | 0.16 |
| iRS < 0 central nonadapting (laser OFF vs. ON) | **Figure 4C** (bottom) | OFF: −0.13 ON: −0.012 | OFF: −0.10 ON: −0.017 | OFF: 0.11 ON: 0.15 | OFF: 0.021 ON: 0.029 | OFF: 0.044 ON: 0.060 | Wilcoxon signed-rank test | V = 30 | 25 | NA | 0.00012 | 0.71 |
| iMM shell nonadapting (laser OFF vs. ON) | **Figure 4D** (top) | OFF: 0.0053 ON: 0.023 | OFF: 0.0062 ON: 0.028 | OFF: 0.13 ON: 0.16 | OFF: 0.0081 ON: 0.010 | OFF: 0.016 ON: 0.020 | Wilcoxon signed-rank test | V = 12,765 | 243 | NA | 0.076 | 0.11 |
| iPE shell nonadapting (laser OFF vs. ON) | **Figure 4D** (middle) | OFF: 0.053 ON: 0.072 | OFF: 0.059 ON: 0.061 | OFF: 0.21 ON: 0.20 | OFF: 0.013 ON: 0.013 | OFF: 0.026 ON: 0.026 | Wilcoxon signed-rank test | V = 13,474 | 243 | NA | 0.22 | 0.079 |
| iRS shell nonadapting (laser OFF vs. ON) | **Figure 4D** (bottom) | OFF: −0.048 ON: −0.049 | OFF: −0.042 ON: −0.041 | OFF: 0.23 ON: 0.22 | OFF: 0.015 ON: 0.014 | OFF: 0.029 ON: 0.028 | Wilcoxon signed-rank test | V = 14,344 | 243 | NA | 0.66 | 0.028 |
| FR change standard central adapting | **Figure 5A** | 2.1 | 2.0 | 5.6 | 0.78 | 1.6 | One-sample t-test | t = 2.7 | 52 | 51 | 0.0092 | 0.38 |
| FR change cascade central adapting | **Figure 5A** | −0.38 | 0.67 | 6.9 | 0.95 | 1.9 | One-sample t-test | t = −0.40 | 52 | 51 | 0.69 | 0.056 |
| FR change deviant central adapting | **Figure 5A** | −2.3 | −2.2 | 5.6 | 0.78 | 1.6 | One-sample t-test | t = −2.9 | 52 | 51 | 0.0054 | 0.40 |
| FR change standard shell adapting | **Figure 5B** | 0.64 | 0.89 | 5.3 | 0.50 | 0.98 | One-sample Wilcoxon test | V = 3760 | 113 | NA | 0.035 | 0.20 |
| FR change cascade shell adapting | **Figure 5B** | 0.50 | 0.44 | 7.3 | 0.68 | 1.4 | One-sample t-test | t = 0.74 | 113 | 112 | 0.46 | 0.069 |
| FR change deviant shell adapting | **Figure 5B** | −1.8 | −1.3 | 7.4 | 0.69 | 1.4 | One-sample Wilcoxon test | V = 2040 | 113 | NA | 0.0057 | 0.26 |
| FR change standard central facilitating | **Figure 5C** | −6.3 | −7.3 | 5.8 | 1.6 | 3.4 | One-sample t-test | t = −4.1 | 14 | 13 | 0.0013 | 1.1 |
| FR change cascade central facilitating | **Figure 5C** | −0.44 | −0.89 | 4.1 | 1.1 | 2.4 | One-sample t-test | t = −0.40 | 14 | 13 | 0.69 | 0.11 |
| FR change deviant central facilitating | **Figure 5C** | 1.5 | 1.3 | 3.4 | 0.92 | 2.0 | One-sample t-test | t = 1.7 | 14 | 13 | 0.12 | 0.45 |
| FR change standard shell facilitating | **Figure 5D** | −2.7 | −3.1 | 5.4 | 0.87 | 1.8 | One-sample t-test | t = −3.1 | 38 | 37 | 0.0042 | 0.50 |
| FR change cascade shell facilitating | **Figure 5D** | 0.36 | 0.44 | 5.1 | 0.84 | 1.7 | One-sample t-test | t = 0.43 | 38 | 37 | 0.67 | 0.070 |
| FR change deviant shell facilitating | **Figure 5D** | 2.6 | 2.7 | 4.5 | 0.74 | 1.5 | One-sample t-test | t = 3.5 | 38 | 37 | 0.0013 | 0.57 |
| FR change standard central nonadapting | **Figure 5E** | −2.5 | −2.2 | 6.2 | 0.50 | 0.99 | One-sample Wilcoxon test | V = 2995 | 155 | NA | 1.4e-06 | 0.38 |
| FR change cascade central nonadapting | **Figure 5E** | −0.68 | −0.44 | 6.3 | 0.51 | 1.0 | One-sample t-test | t = −1.3 | 155 | 154 | 0.18 | 0.11 |
| FR change deviant central nonadapting | **Figure 5E** | 0.57 | 0.0 | 5.8 | 0.47 | 0.93 | One-sample t-test | t = 1.2 | 155 | 154 | 0.22 | 0.098 |
| FR change standard shell nonadapting | **Figure 5F** | −0.63 | −0.44 | 5.3 | 0.34 | 0.68 | One-sample Wilcoxon test | V = 11,050 | 243 | NA | 0.035 | 0.14 |
| FR change cascade shell nonadapting | **Figure 5F** | −0.51 | −0.44 | 5.1 | 0.32 | 0.64 | One-sample Wilcoxon test | V = 12,157 | 243 | NA | 0.15 | 0.089 |
| FR change deviant shell nonadapting | **Figure 5F** | −0.059 | 0.0 | 5.0 | 0.32 | 0.64 | One-sample t-test | t = −0.18 | 243 | 242 | 0.86 | 0.012 |
| FR central facilitating (first vs. last standard) | **Figure 6C** | First: 31 Last: 36 | First: 29 Last: 31 | First: 15 Last: 16 | First: 3.9 Last: 4.4 | First: 8.5 Last: 9.5 | Wilcoxon signed-rank test | V = 0 | 14 | NA | 0.0017 | 0.87 |

*Table 1 continued on next page*

## Table 1 continued

| Comparison | Figure | Mean | Median | SD | SEM | CI (±) | Test | Test statistic | N | df | p | Effect size |
|---|---|---|---|---|---|---|---|---|---|---|---|---|
| FR shell facilitating (first vs. last standard) | Figure 6D | First: 53 Last: 57 | First: 38 Last: 42 | First: 38 Last: 42 | First: 6.2 Last: 6.8 | First: 13 Last: 14 | Wilcoxon signed-rank test | V = 92 | 38 | NA | 9.3e-05 | 0.64 |
| FR central adapting (cascade vs. many standards) | Figure 3—figure supplement 2B (left) | Casc: 61 MS: 63 | Casc: 50 MS: 52 | Casc: 38 MS: 40 | Casc: 5.2 MS: 5.6 | Casc: 10 MS: 11 | Wilcoxon signed-rank test | V = 595 | 52 | NA | 0.39 | 0.12 |
| FR central facilitating (cascade vs. many standards) | Figure 3—figure supplement 2B (right) | Casc: 29 MS: 31 | Casc: 26 MS: 28 | Casc: 14 MS: 16 | Casc: 3.8 MS: 4.3 | Casc: 8.2 MS: 9.3 | Wilcoxon signed-rank test | V = 41 | 14 | NA | 0.49 | 0.19 |
| FR shell adapting (cascade vs. many standards) | Figure 3—figure supplement 2C (left) | Casc: 64 MS: 66 | Casc: 43 MS: 41 | Casc: 61 MS: 68 | Casc: 5.7 MS: 6.4 | Casc: 11 MS: 13 | Wilcoxon signed-rank test | V = 2653 | 113 | NA | 0.46 | 0.064 |
| FR shell facilitating (cascade vs. many standards) | Figure 3—figure supplement 2C (right) | Casc: 43 MS: 45 | Casc: 24 MS: 28 | Casc: 41 MS: 52 | Casc: 6.6 MS: 8.4 | Casc: 13 MS: 17 | Wilcoxon signed-rank test | V = 264.5 | 38 | NA | 0.41 | 0.14 |
| Central iMM OFF (single vs. multiunit) | Figure 3—figure supplement 3 (left) | Single: 0.045 Multi: 0.057 | Single: 0.048 Multi: 0.064 | Single: 0.15 Multi: 0.18 | Single: 0.052 Multi: 0.013 | Single: 0.12 Multi: 0.025 | Wilcoxon rank-sum test | W = 825 | Single: 8 Multi: 213 | NA | 0.88 | 0.010 |
| Central iMM ON (single vs. multiunit) | Figure 3—figure supplement 3 (left) | Single: 0.087 Multi: 0.092 | Single: 0.085 Multi: 0.086 | Single: 0.17 Multi: 0.16 | Single: 0.059 Multi: 0.011 | Single: 0.14 Multi: 0.022 | Student's t-test | t = −0.093 | Single: 8 Multi: 213 | 7.5 | 0.93 | 0.034 |
| Shell iMM OFF (single vs. multiunit) | Figure 3—figure supplement 3 (right) | Single: 0.035 Multi: 0.081 | Single: 0.028 Multi: 0.055 | Single: 0.18 Multi: 0.25 | Single: 0.022 Multi: 0.014 | Single: 0.045 Multi: 0.027 | Wilcoxon rank-sum test | W = 9832 | Single: 67 Multi: 327 | NA | 0.19 | 0.067 |
| Shell iMM ON (single vs. multiunit) | Figure 3—figure supplement 3 (right) | Single: 0.046 Multi: 0.091 | Single: 0.045 Multi: 0.072 | Single: 0.21 Multi: 0.23 | Single: 0.026 Multi: 0.013 | Single: 0.051 Multi: 0.025 | Wilcoxon rank-sum test | W = 9883 | Single: 67 Multi: 327 | NA | 0.21 | 0.064 |

iRS: index of repetition suppression; iPE: index of prediction error; iMM: index of neuronal mismatch; Aw: awake; An: anesthetized; casc: cascading; MS: many standards.

## IC units encode different aspects of prediction and repetition in awake and anesthetized states

Much of the research regarding SSA and deviance detection in IC to date has been performed in anesthetized animals, with few studies recording from awake subjects (*Duque and Malmierca, 2015*; *Parras et al., 2017*). Given that neuronal responses to sound depend on the state of anesthesia of the subject, it is possible that there are differences in predictive coding metrics between the awake and anesthetized states (*Fontanini and Katz, 2008*; *Gaese and Ostwald, 2001*; *Schumacher et al., 2011*). While previous studies have characterized how anesthesia affects SSA, it remains unknown whether its component repetition and prediction metrics differ with anesthetic state (*Duque and Malmierca, 2015*). Therefore, we first characterized how anesthesia affects these predictive coding metrics in a subset of animals. We first performed awake recordings and then repeated our experimental procedures, leaving the animal head-fixed and the probe in place, after anesthetizing the mouse with isoflurane (*Figure 2A*). This protocol allowed us to compare how metrics of predictive coding differ between the awake and anesthetized preparations in the same population of units.

In the central IC, the mean iMM in the anesthetized condition was positive, indicative of prevalent adaptation (*Figure 2B*). The iMM values under anesthesia were significantly higher than those obtained while the animal was awake (*Figure 2B*, *Table 1*; p=8.8e-05, Wilcoxon rank-sum test). To better understand what prediction or repetition effects underlie iMM in each condition, the iMM for both distributions was further decomposed into an iPE and iRS. In the anesthetized condition, the mean iPE value of 0.077 indicated the presence of modest prediction error, while a mean iPE of −0.13 indicated that negative prediction error is significantly more prevalent in the awake condition (*Figure 2C*, *Table 1*; p=0.017, Student's *t*-test). Under both anesthetized and awake conditions, prominent repetition suppression was observed in the central IC (*Figure 2D*).

Similar to the central IC, the mean iMM was significantly more positive in shell regions during anesthesia (*Figure 2E*, *Table 1*; p=3.5e-08, Wilcoxon rank-sum test). A greater proportion of units in the awake condition had a negative iMM compared with the anesthetized distribution, indicating that facilitation (a greater response to the standard than the deviant context) is more common in the awake than the anesthetized condition (*Figure 2E*). The iPE values in shell IC suggest that prediction error is significantly higher in the awake compared to the anesthetized condition (*Figure 2F*, *Table 1*; p=2.6e-05, Wilcoxon rank-sum test). Although the distribution for the iRS under anesthesia had a positive mean of 0.25, indicating prevalent repetition suppression, the awake distribution exhibited a significant leftward shift by comparison (*Figure 2G*). Interestingly, the mean iRS for the awake condition was negative (mean = −0.056), indicating that repetition *enhancement*, rather than suppression, is present in the awake shell IC (*Figure 2G*, *Table 1*; p=2.5e-16, Wilcoxon rank-sum test). These results point to differences between predictive coding metrics in the awake and anesthetized states, with previously undescribed metrics such as repetition enhancement and negative prediction error more prominent in awake animals.

## Adapting and facilitating units are differentially affected by cortico-collicular inactivation

We next performed recordings in IC of awake mice to determine how neuronal mismatch and its component repetition and prediction metrics were affected by cortico-collicular inactivation (*Figure 3A*). To inactivate cortico-collicular feedback, we shined light over AC in subjects that expressed a suppressive opsin in cortico-collicular neurons. We segregated the population of recorded units according to those that exhibited a significantly stronger response to the deviant than the standard (adapting units; *Figure 3B*, blue; Figure 5C), those that exhibited a significantly stronger response to the standard than the deviant (facilitating units; *Figure 3B*, red; Figure 5F), and those that responded equally to both stimulus contexts (nonadapting units; *Figure 3B*, green) for recordings in both central and shell regions of IC (*Figure 3B*, left vs. right).

The iMM for adapting units in the central nucleus significantly decreased with laser inactivation of cortico-collicular neurons (*Figure 3D*, top; *Table 1*; p=0.00034, Wilcoxon signed-rank test). The iMM at baseline for adapting units predominantly represents repetition suppression (*Figure 3D*, bottom) and a small amount of prediction error (*Figure 3D*, middle). Prediction error was abolished during laser inactivation (*Figure 3D*, middle; *Table 1*; p=0.048, Wilcoxon signed-rank test), while repetition suppression remained unaffected (*Figure 3D*, bottom). Adapting units in shell regions of IC exhibited

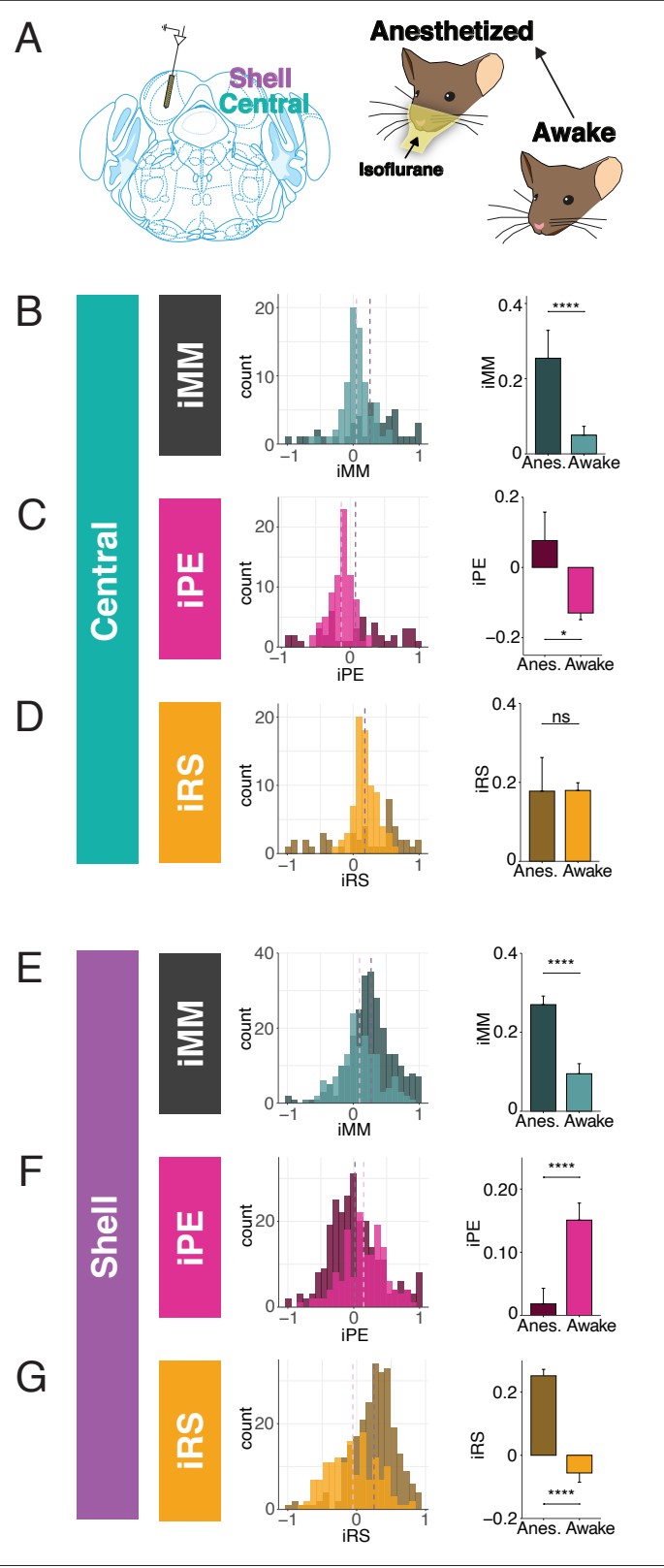

**Figure 2.** Inferior colliculus (IC) units encode different aspects of prediction and repetition in awake and anesthetized states. (**A**) Experimental design for recording in the awake and isoflurane anesthetized IC in the same population of units. (**B**) Distribution of index of neuronal mismatch (iMM) in the awake vs. anesthetized central IC. Bar plots represent means over the population of n = 39 units. Error bars are standard error of the mean. (**C**) Index

*Figure 2 continued on next page*

*Figure 2 continued*

of prediction error (iPE) distribution in the awake vs. anesthetized central IC. (**D**) Index of repetition suppression (iRS) distribution in the awake vs. anesthetized central IC. (**E**) Distribution of iMM in the awake vs. anesthetized shell IC. Bar plots represent means over the population of n = 165 units. Error bars are standard error of the mean. (**F**) iPE distribution in the awake vs. anesthetized shell IC. (**G**) iRS distribution in the awake vs. anesthetized shell IC. Data is from four recording sessions in one mouse.

a similar pattern to those in the central nucleus. At baseline, these units encoded both prediction error and repetition suppression (*Figure 3E*, middle and bottom). A significant decrease in iMM during laser inactivation (*Figure 3E*, top; *Table 1*; p=0.0023, Wilcoxon signed-rank test) was driven by a decrease in prediction error (*Figure 3E*, middle; *Table 1*; p=0.034, Wilcoxon signed-rank test), whereas repetition suppression remained unaffected (*Figure 3E*, bottom). Combined, these results suggest that removing cortical feedback reduced prediction error but not repetition suppression in adapting units.

Prior studies of deviance detection in IC have focused exclusively on adapting units. However, given the relative prevalence of facilitating units discovered in the awake versus anesthetized IC (*Figure 2*), we further investigated this population of units to determine whether facilitation reflects prediction or repetition effects. In the central nucleus, cortico-collicular inactivation led to a significant decrease in facilitation in facilitating units (*Figure 3G*, top; *Table 1*; p=0.0036, Student's *t*-test). At baseline, the iMM for facilitating units represents a combination of negative prediction error and repetition enhancement (*Figure 3G*, middle and bottom). During inactivation, negative prediction error remained unaffected (*Figure 3G*, middle), while repetition enhancement was nearly abolished (*Figure 3G*, bottom; *Table 1*; p=0.0026, Student's *t*-test). Facilitating units in the shell IC were also significantly affected by cortico-collicular inactivation (*Figure 3H*, top; *Table 1*; p=0.0016, Wilcoxon signed-rank test). In this case, however, the change in iMM was driven by the near abolishment of negative prediction error (*Figure 3H*, middle; *Table 1*; p=0.037, Wilcoxon signed-rank test), while repetition enhancement was unaffected (*Figure 3H*, bottom).

These data suggest that adaptation and facilitation in the awake IC are composed of distinct underlying processes: adapting populations in both central and shell regions of IC exhibit prediction error and repetition suppression, while facilitating populations are characterized by negative prediction error and repetition enhancement. In adapting units in both central and shell regions, cortico-collicular inactivation significantly decreases prediction error. Facilitating units in the central IC display decreased repetition enhancement with cortico-collicular inactivation, while those in shell regions exhibit decreased negative prediction error. To ensure that the laser-induced changes described above were opsin-mediated, we performed control experiments in two mice with identical manipulations to the experimental group, but in the absence of ArchT (*Figure 3—figure supplement 1A*). At baseline, the control group exhibited a similar distribution of iMM values to the experimental group in both the central and shell regions of IC (*Figure 3—figure supplement 1B*, *Table 2*). Similar proportions of adapting/facilitating/nonadapting units were also found in the control (central: 23% adapting, 5% facilitating, 71% nonadapting; shell: 29% adapting, 18% facilitating, 53% nonadapting) and experimental groups (central: 24% adapting, 6% facilitating, 70% nonadapting; shell: 29% adapting, 9% facilitating, 62% nonadapting). We found no significant differences between baseline and laser trials for either adapting (*Figure 3—figure supplement 1C and D*, *Table 2*) or facilitating (*Figure 3—figure supplement 1E and F*) units in either region. This experiment confirmed that the observed effects of cortico-collicular inactivation were indeed due to opsin-mediated inactivation of the cortico-collicular projection neurons.

## Adapting and facilitating units respond similarly to the cascade and many standards controls

Though the cascade sequence is free of repetition effects between adjacent tone pairs, it does exhibit global repetition across the entire tone sequence. To assess whether global stimulus regularity affects the response to the cascade context, we used a shuffled version of the cascade sequence, known as the 'many standards' sequence, as an additional control stimulus (*Figure 3—figure supplement 2A*). The many standards sequence contains the same 10 tones as the cascade but presented in random order (*Figure 3—figure supplement 2A*). This reduces the potential for adaptation across adjacent frequency channels and also eliminates the global predictability of the stimulus, both of which could

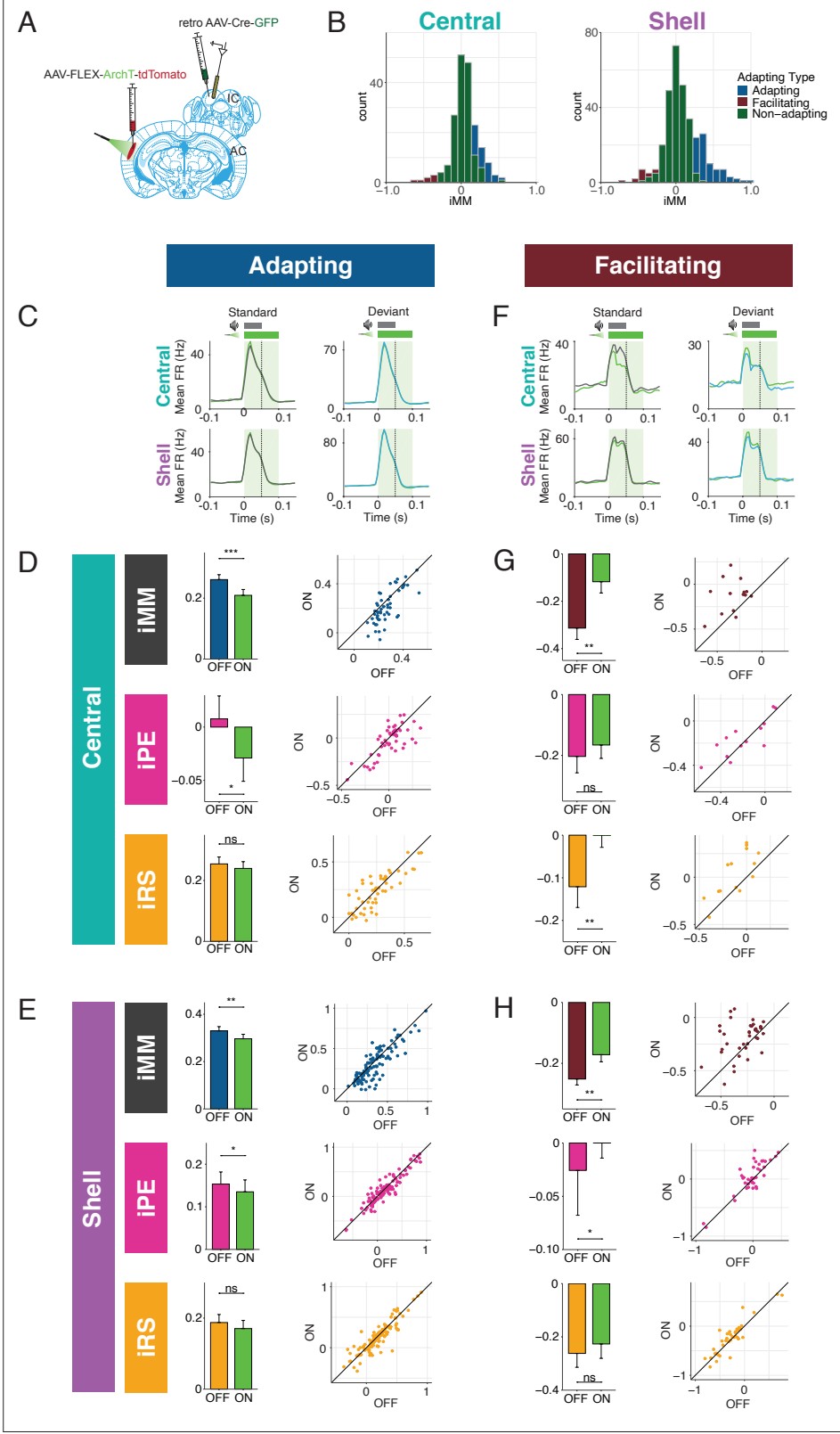

**Figure 3.** Adapting and facilitating inferior colliculus (IC) units are differentially affected by cortico-collicular inactivation. (**A**) Experimental design for recording in awake IC during laser inactivation of the cortico-collicular pathway. (**B**) Categorization of units according to whether they displayed significant adaptation, facilitation, or neither (nonadapting). (**C**) Average peristimulus time histogram for adapting units in central (top) and shell

*Figure 3 continued*

(bottom) IC. Green = during laser inactivation. (**D**) Index of neuronal mismatch (iMM) (top), index of prediction error (iPE) (middle), and index of repetition suppression (iRS) (bottom) for adapting units in the central nucleus. Dots represent recorded units. Bar plots represent means over the population of n = 52 units. Error bars are standard error of the mean. (**E**) iMM (top), iPE (middle), and iRS (bottom) for adapting units in shell regions of IC. Dots represent recorded units. Bar plots represent means over the population of n = 113 units. Error bars are standard error of the mean. (**F**) Average peristimulus time histogram for facilitating units in central (top) and shell (bottom) IC. Green = during laser inactivation. (**G**) iMM (top), iPE (middle), and iRS (bottom) for facilitating units in the central nucleus. Dots represent recorded units. Bar plots represent means over the population of n = 14 units. Error bars are standard error of the mean. (**H**) iMM (top), iPE (middle), and iRS (bottom) for facilitating units in shell regions of IC. Dots represent recorded units. Bar plots represent means over the population of n = 38 units. Error bars are standard error of the mean.

The online version of this article includes the following figure supplement(s) for figure 3:

**Figure supplement 1.** Control data.

**Figure supplement 2.** Comparison of neuronal responses between the many standards and cascade sequences.

**Figure supplement 3.** Index of neuronal mismatch (iMM) distribution does not differ between single- and multiunit types.

lead to suppression of responses to tones in the cascade context and potentially affect the calculations of iMM, iPE, and iRS. We compared the responses of adapting and facilitating units in both central and shell regions of IC to tones in the cascade versus the many standards context (***Figure 3—figure supplement 2A***). We found no significant differences in firing rates to the cascade versus the many standards contexts (***Figure 3—figure supplement 2B and C***, ***Table 1***), suggesting that the global structure of the cascade sequence does not significantly affect how units in IC respond to this stimulus, as has been shown in other structures (***Casado-Román et al., 2020***; ***Parras et al., 2021***).

## iMM distribution does not differ between single- and multiunit types

The analysis of changes in predictive coding metrics is performed on pooled single- and multiunit responses of IC units. To determine whether the expression of neuronal mismatch differs between these unit types, we plotted the iMM for laser OFF and ON conditions for each of the subgroups in the central and shell regions of the IC separated by single- (displayed in teal) and multiunits (***Figure 3—figure supplement 3***). We observed no differences in the distributions of these unit types in central or shell IC (***Table 1***; central OFF: p=0.88, Wilcoxon rank-sum test; central ON: p=0.93, Student's *t*-test; shell OFF: p=0.19, Wilcoxon rank-sum test; shell ON: p=0.21, Wilcoxon rank-sum test). We therefore combined data from both single- and multiunits for the analyses of predictive coding metrics.

## Nonadapting units also display top-down repetition enhancement

The majority of units in both central and shell IC do not exhibit either adaptation or facilitation but respond similarly to tones when they are presented as a standard or deviant (***Figure 4A***). However, since both negative and positive metrics are included in the calculation of iMM, it is still possible that these units exhibit predictive processing that may not be reflected in the overall iMM value. We further characterized these nonadapting units (***Figure 4B***) and tested how they are affected by cortico-collicular inactivation. Nonadapting units in the central nucleus exhibited a significant increase in iMM during inactivation (***Figure 4C***, top; ***Table 1***; p=2.7e-06, Wilcoxon signed-rank test), whereas those in the shell IC were unaffected (***Figure 4D***, top). The change in iMM for nonadapting units in the central nucleus was driven by a significant increase in iRS (***Figure 4C***, bottom middle; ***Table 1***; p=0.0011, Wilcoxon signed-rank test). To determine whether this reflected a change in repetition suppression or enhancement, we further segregated central nonadapting units according to whether their baseline iRS values were negative or positive (***Figure 4C***, bottom). Only those units with negative baseline iRS values (i.e., those units showing repetition enhancement) were significantly affected by cortico-collicular inactivation (***Figure 4C***, bottom; ***Table 1***; p=0.00012, Wilcoxon signed-rank test). In control experiments without ArchT, no significant changes were observed in nonadapting units (***Figure 3—figure supplement 1G and H***, ***Table 2***). These results indicate that, similar to central facilitating units, central nonadapting units display repetition enhancement, and that input from the cortex is critical for expression of this phenomenon.

**Table 2.** Statistical comparisons for control data.

| Comparison | Figure | Mean | Median | SD | SEM | CI (±) | Test | Test statistic | N | df | p | Effect size |
|---|---|---|---|---|---|---|---|---|---|---|---|---|
| iMM central (control vs. experimental) | *Figure 3—figure supplement* 1B (left) | Con: 0.092 Exp: 0.057 | Con: 0.086 Exp: 0.064 | Con: 0.16 Exp: 0.18 | Con: 0.011 Exp: 0.012 | Con: 0.022 Exp: 0.024 | Wilcoxon rank-sum test | W = 7919 | 77 (control) 221 (exp.) | NA | 0.37 | 0.052 |
| iMM shell (control vs. experimental) | *Figure 3—figure supplement* 1B (right) | Con: 0.083 Exp: 0.073 | Con: 0.069 Exp: 0.053 | Con: 0.23 Exp: 0.24 | Con: 0.012 Exp: 0.012 | Con: 0.023 Exp: 0.024 | Wilcoxon rank-sum test | W = 22,364 | 119 (control) 394 (exp.) | NA | 0.45 | 0.034 |
| iMM central adapting (laser OFF vs. ON) | *Figure 3—figure supplement* 1C (top) | OFF: 0.35 ON: 0.33 | OFF: 0.35 ON: 0.32 | OFF: 0.11 ON: 0.15 | OFF: 0.026 ON: 0.034 | OFF: 0.054 ON: 0.072 | Wilcoxon signed-rank test | V = 124 | 18 | NA | 0.099 | 0.40 |
| iPE central adapting (laser OFF vs. ON) | *Figure 3—figure supplement* 1C (middle) | OFF: 0.16 ON: 0.19 | OFF: 0.10 ON: 0.081 | OFF: 0.39 ON: 0.40 | OFF: 0.091 ON: 0.094 | OFF: 0.19 ON: 0.20 | Paired t-test | t = −1.1 | 18 | 17 | 0.30 | 0.25 |
| iRS central adapting (laser OFF vs. ON) | *Figure 3—figure supplement* 1C (bottom) | OFF: 0.19 ON: 0.14 | OFF: 0.24 ON: 0.14 | OFF: 0.38 ON: 0.37 | OFF: 0.090 ON: 0.087 | OFF: 0.19 ON: 0.18 | Paired t-test | t = 1.9 | 18 | 17 | 0.077 | 0.44 |
| iMM shell adapting (laser OFF vs. ON) | *Figure 3—figure supplement* 1D (top) | OFF: 0.38 ON: 0.38 | OFF: 0.35 ON: 0.38 | OFF: 0.19 ON: 0.22 | OFF: 0.032 ON: 0.037 | OFF: 0.065 ON: 0.075 | Paired t-test | t = −0.0013 | 35 | 34 | 0.99 | 0.00022 |
| iPE shell adapting (laser OFF vs. ON) | *Figure 3—figure supplement* 1D (middle) | OFF: 0.16 ON: 0.14 | OFF: 0.12 ON: 0.15 | OFF: 0.24 ON: 0.26 | OFF: 0.041 ON: 0.044 | OFF: 0.083 ON: 0.090 | Paired t-test | t = 0.58 | 35 | 34 | 0.56 | 0.099 |
| iRS shell adapting (laser OFF vs. ON) | *Figure 3—figure supplement* 1D (bottom) | OFF: 0.22 ON: 0.24 | OFF: 0.24 ON: 0.20 | OFF: 0.23 ON: 0.22 | OFF: 0.040 ON: 0.038 | OFF: 0.081 ON: 0.077 | Paired t-test | t = −0.78 | 35 | 34 | 0.44 | 0.13 |
| iMM central facilitating (laser OFF vs. ON) | *Figure 3—figure supplement* 1E (top) | OFF: −0.37 ON: −0.33 | OFF: −0.36 ON: −0.37 | OFF: 0.15 ON: 0.18 | OFF: 0.077 ON: 0.090 | OFF: 0.25 ON: 0.29 | Paired t-test | t = −1.1 | 4 | 3 | 0.34 | 0.57 |
| iPE central facilitating (laser OFF vs. ON) | *Figure 3—figure supplement* 1E (middle) | OFF: −0.043 ON: 0.030 | OFF: −0.0047 ON: 0.077 | OFF: 0.47 ON: 0.45 | OFF: 0.24 ON: 0.22 | OFF: 0.75 ON: 0.71 | Paired t-test | t = −0.93 | 4 | 3 | 0.42 | 0.47 |
| iRS central facilitating (laser OFF vs. ON) | *Figure 3—figure supplement* 1E (bottom) | OFF: −0.33 ON: −0.36 | OFF: −0.49 ON: −0.53 | OFF: 0.55 ON: 0.60 | OFF: 0.27 ON: 0.30 | OFF: 0.87 ON: 0.95 | Paired t-test | t = 0.49 | 4 | 3 | 0.66 | 0.24 |
| iMM shell facilitating (laser OFF vs. ON) | *Figure 3—figure supplement* 1F (top) | OFF: −0.38 ON: −0.31 | OFF: −0.32 ON: −0.30 | OFF: 0.22 ON: 0.20 | OFF: 0.048 ON: 0.043 | OFF: 0.10 ON: 0.090 | Wilcoxon signed-rank test | V = 63 | 21 | NA | 0.070 | 0.40 |
| iPE shell facilitating (laser OFF vs. ON) | *Figure 3—figure supplement* 1F (middle) | OFF: −0.090 ON: −0.094 | OFF: −0.11 ON: −0.081 | OFF: 0.18 ON: 0.20 | OFF: 0.040 ON: 0.044 | OFF: 0.083 ON: 0.093 | Wilcoxon signed-rank test | V = 109 | 21 | NA | 0.84 | 0.050 |
| iRS shell facilitating (laser OFF vs. ON) | *Figure 3—figure supplement* 1F (bottom) | OFF: −0.29 ON: −0.21 | OFF: −0.28 ON: −0.15 | OFF: 0.24 ON: 0.21 | OFF: 0.053 ON: 0.047 | OFF: 0.11 ON: 0.097 | Paired t-test | t = −1.8 | 21 | 20 | 0.091 | 0.39 |
| iMM central nonadapting (laser OFF vs. ON) | *Figure 3—figure supplement* 1G (top) | OFF: 0.021 ON: 0.060 | OFF: 0.014 ON: 0.050 | OFF: 0.24 ON: 0.23 | OFF: 0.032 ON: 0.031 | OFF: 0.064 ON: 0.063 | Paired t-test | t = −1.8 | 55 | 54 | 0.075 | 0.24 |
| iPE central nonadapting (laser OFF vs. ON) | *Figure 3—figure supplement* 1G (middle) | OFF: 0.12 ON: 0.14 | OFF: 0.034 ON: 0.092 | OFF: 0.34 ON: 0.35 | OFF: 0.046 ON: 0.047 | OFF: 0.092 ON: 0.095 | Paired t-test | t = −1.2 | 55 | 54 | 0.23 | 0.16 |
| iRS central nonadapting (laser OFF vs. ON) | *Figure 3—figure supplement* 1G (bottom) | OFF: −0.095 ON: −0.083 | OFF: −0.064 ON: −0.072 | OFF: 0.31 ON: 0.29 | OFF: 0.042 ON: 0.038 | OFF: 0.084 ON: 0.077 | Paired t-test | t = −0.57 | 55 | 54 | 0.57 | 0.077 |
| iMM shell nonadapting (laser OFF vs. ON) | *Figure 3—figure supplement* 1H (top) | OFF: 0.063 ON: 0.051 | OFF: 0.040 ON: 0.031 | OFF: 0.16 ON: 0.22 | OFF: 0.021 ON: 0.027 | OFF: 0.042 ON: 0.054 | Wilcoxon signed-rank test | V = 1133 | 63 | NA | 0.39 | 0.11 |
| iPE shell nonadapting (laser OFF vs. ON) | *Figure 3—figure supplement* 1H (middle) | OFF: 0.053 ON: 0.027 | OFF: 0.0 ON: 0.0 | OFF: 0.25 ON: 0.26 | OFF: 0.031 ON: 0.032 | OFF: 0.063 ON: 0.065 | Paired t-test | t = 0.88 | 63 | 62 | 0.38 | 0.11 |
| iRS shell nonadapting (laser OFF vs. ON) | *Figure 3—figure supplement* 1H (bottom) | OFF: 0.011 ON: 0.024 | OFF: 0.028 ON: 0.041 | OFF: 0.27 ON: 0.28 | OFF: 0.034 ON: 0.035 | OFF: 0.068 ON: 0.071 | Paired t-test | t = −0.43 | 63 | 62 | 0.67 | 0.054 |
| iRS > 0 central nonadapting (laser OFF vs. ON) | N/A | OFF: 0.21 ON: 0.18 | OFF: 0.20 ON: 0.16 | OFF: 0.12 ON: 0.16 | OFF: 0.026 ON: 0.034 | OFF: 0.054 ON: 0.070 | Paired t-test | t = 1.5 | 22 | 21 | 0.16 | 0.31 |
| iRS < 0 central nonadapting (laser OFF vs. ON) | N/A | OFF: −0.31 ON: −0.26 | OFF: −0.27 ON: −0.27 | OFF: 0.21 ON: 0.21 | OFF: 0.036 ON: 0.037 | OFF: 0.074 ON: 0.075 | Paired t-test | t = −1.7 | 32 | 31 | 0.099 | 0.30 |

iRS: index of repetition suppression; iPE: index of prediction error; iMM: index of neuronal mismatch.

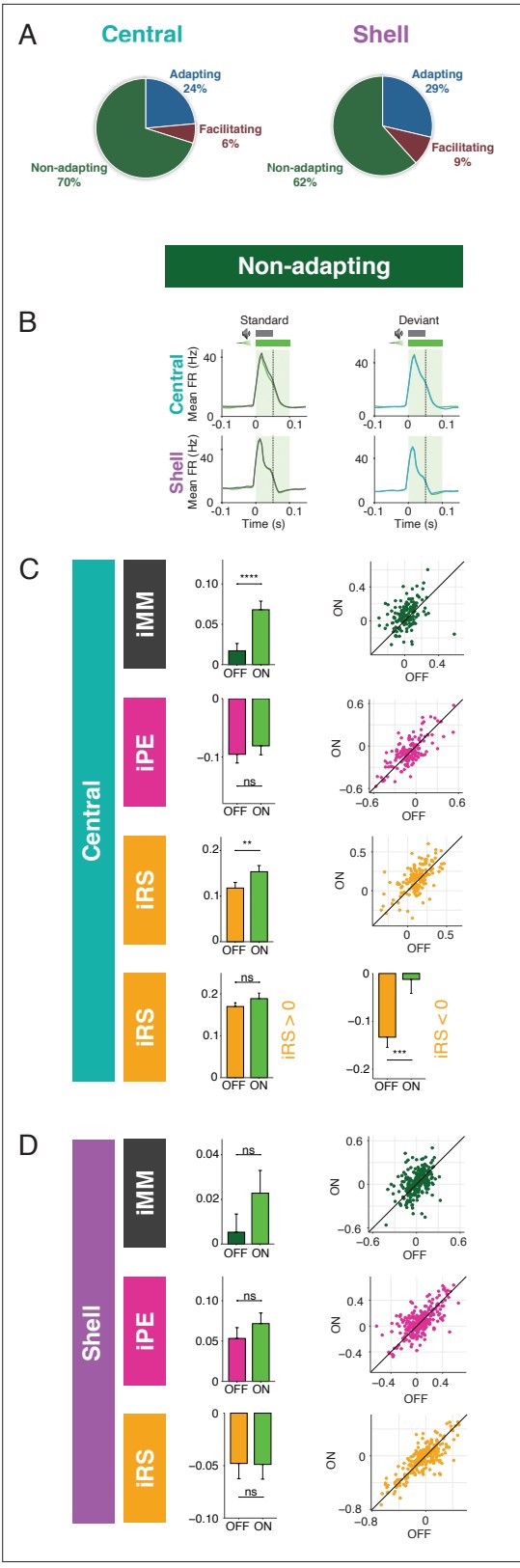

**Figure 4.** Nonadapting units also display top-down repetition enhancement. (**A**) Distribution of adapting types (adapting, facilitating, and nonadapting) for units in central (left) and shell (right) regions of the inferior colliculus (IC). (**B**) Average peristimulus time histogram for nonadapting units in central (top) and shell (bottom) IC. (**C**) Index of neuronal mismatch (iMM) (top), index of prediction error (iPE) (middle), and index of repetition suppression (iRS)

*Figure 4 continued on next page*

*Figure 4 continued*

(bottom) for nonadapting units in central regions of IC. Dots represent recorded units. Bar plots represent means over the population of n = 155 units. Error bars are standard error of the mean. (**D**) iMM (top), iPE (middle), and iRS (bottom) for nonadapting units in shell regions of IC. Dots represent recorded units. Bar plots represent means over the population of n = 243 units. Error bars are standard error of the mean.

## Standard and deviant responses are bidirectionally modulated by cortico-collicular inactivation

The observed changes in repetition metrics with cortico-collicular inactivation could reflect an effect on either the standard or cascade context. Similarly, the shift in prediction metrics observed with inactivation could be due to altered responses to either the cascade or deviant contexts. We next determined whether the laser-induced changes in the iMM, iPE, and iRS for adapting units reflect changes in the firing rates to the standard, deviant, or cascade contexts. We found that adapting units in the central nucleus increased responses to the standard (*Figure 5A*, *Table 1*; p=0.0092, one-sample *t*-test) and decreased responses to the deviant (*Figure 5A*, *Table 1*; p=0.0054, one-sample *t*-test) during inactivation. These results explain the decrease in iMM for this population during the laser stimulus (*Figure 3D*, top): the firing rate to the cascade stimulus did not change during cortico-collicular inactivation, which means that the decrease in firing rate to the deviant alone underlies the decrease in prediction error observed for this population (*Figure 3D*, middle). Adapting units in the shell exhibited the same pattern of bidirectional changes to the standard (*Figure 5B*, *Table 1*; p=0.035, one-sample Wilcoxon test) and deviant (*Figure 5B*, *Table 1*; p=0.0057, one-sample Wilcoxon test), similarly accounting for their decrease in iMM and prediction error (*Figure 3E*), with no change in response to the cascade condition (*Figure 5B*). These data suggest that inactivation of the cortico-collicular pathway induces bidirectional changes in firing rates to the standard and deviant for adapting units in both central and shell regions of IC.

We also investigated how responses to each stimulus context changed with cortico-collicular inactivation for facilitating units. For central facilitating units, only the firing rate to the standard context changed during inactivation (*Figure 5C*, *Table 1*; p=0.0013, one-sample *t*-test), explaining the observed change in repetition enhancement for this population (*Figure 3G*). For shell facilitating units, a decreased response to the standard (*Figure 5D*, *Table 1*; p=0.0042, one-sample *t*-test) and an increased response to the deviant (*Figure 5D*, *Table 1*; p=0.0013, one-sample *t*-test) were elicited on laser trials, accounting for changes in the iMM and the abolishment of negative prediction error (*Figure 3H*). These changes are directionally opposite to the observed firing rate changes observed for adapting units under inactivation, with a decrease to the standard context for both central and shell units and an increase to the deviant context for shell units.

For nonadapting units, a significant decrease in response to the standard context was observed in both central (*Figure 5E*, *Table 1*; p=1.4e-06, one-sample Wilcoxon test) and shell (*Figure 5F*, *Table 1*; p=0.035, one-sample Wilcoxon test) regions of IC. The decrease was only significant enough to produce an effect on the iMM in central regions (*Figure 4C*, top), leading to an increase in repetition suppression (*Figure 4C*, bottom).

For adapting and facilitating units, these data exhibit that IC responses to the standard and deviant contexts in the absence of cortical input are bidirectionally modulated, such that neurons respond more similarly to both contexts rather than firing differentially to each. For nonadapting units, the response to the standard context alone is diminished during cortico-collicular inactivation, causing these units to become more adapting. These changes suggest that under normal conditions AC provides information regarding sound context to neurons in IC.

## IC units have distinct combinations of iPE and iRS

To determine whether IC units exhibit particular combinations of repetition suppression/enhancement and prediction error/negative prediction error, we plotted the iPE values against the iRS values for each unit in the adapting, facilitating, and nonadapting groups. Both the adapting and nonadapting groups in the central IC contained units with significant values for both iPE and iRS, most often resulting from a combination of negative prediction error and repetition suppression (*Figure 6A*, maroon dots). In the shell IC, a greater variety of response combinations was observed. All three groups contained units with both significant negative prediction error and repetition suppression, as well as a separate

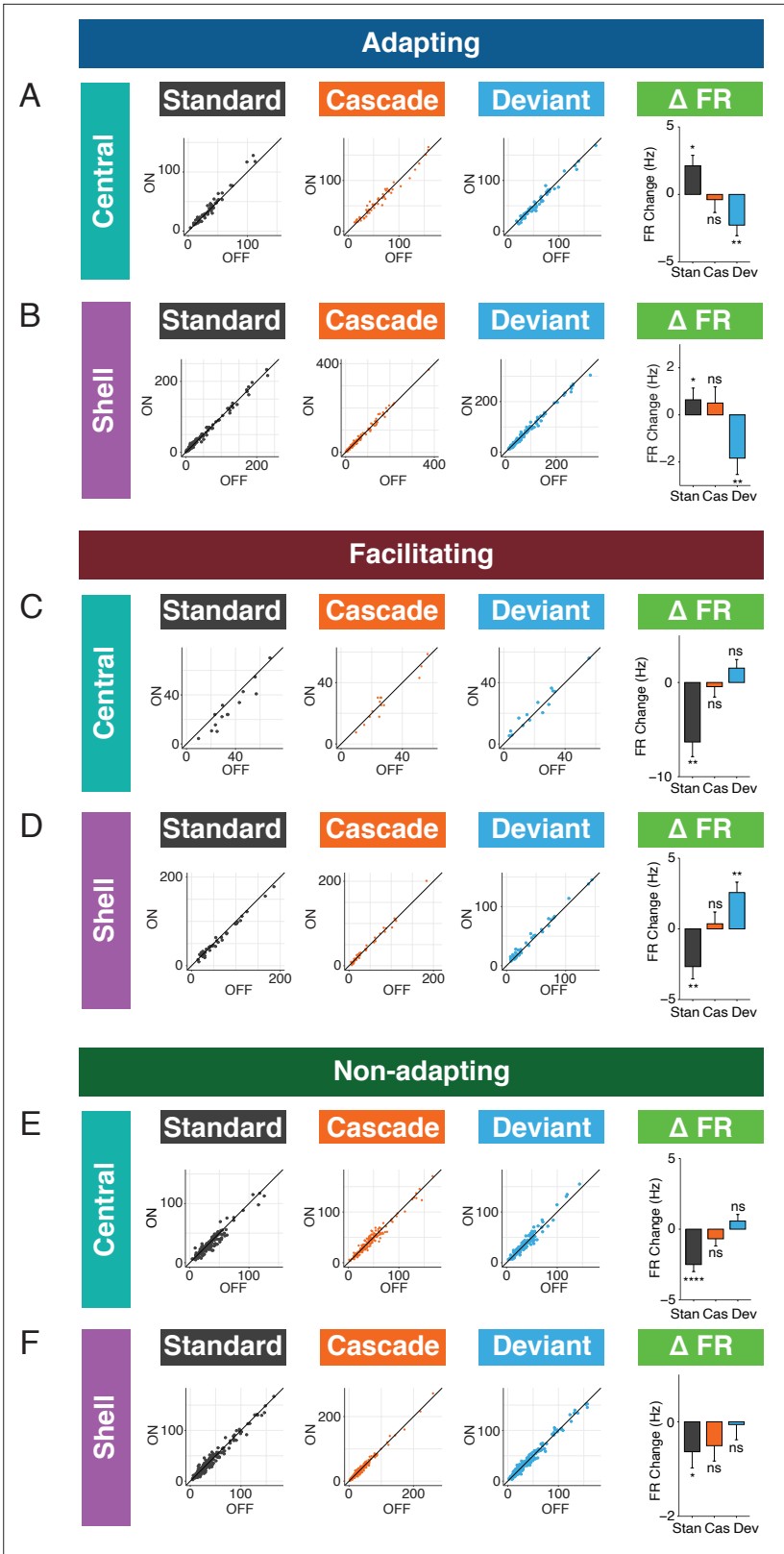

**Figure 5.** Standard and deviant responses are bidirectionally modulated by cortico-collicular inactivation. (**A**) Responses to the standard (left), cascade (middle left), and deviant (middle right) for adapting units in central regions of the inferior colliculus (IC) under baseline and laser conditions. Change in firing rate between the laser and baseline condition for each stimulus (right). Dots represent recorded units. Bar plots represent means over

*Figure 5 continued on next page*

*Figure 5 continued*

the population of n = 52 units. Error bars are standard error of the mean. (**B**) Responses to the standard (left), cascade (middle left), and deviant (middle right) for adapting units in shell regions of IC under baseline and laser conditions. Change in firing rate between the laser and baseline condition for each stimulus (right). Dots represent recorded units. Bar plots represent means over the population of n = 113 units. Error bars are standard error of the mean. (**C**) Responses to the standard (left), cascade (middle left), and deviant (middle right) for facilitating units in central regions of IC under baseline and laser conditions. Change in firing rate between the laser and baseline condition for each stimulus (right). Dots represent recorded units. Bar plots represent means over the population of n = 14 units. Error bars are standard error of the mean. (**D**) Responses to the standard (left), cascade (middle left), and deviant (middle right) for facilitating units in shell regions of IC under baseline and laser conditions. Change in firing rate between the laser and baseline condition for each stimulus (right). Dots represent recorded units. Bar plots represent means over the population of n = 38 units. Error bars are standard error of the mean. (**E**) Responses to the standard (left), cascade (middle left), and deviant (middle right) for nonadapting units in central regions of IC under baseline and laser conditions. Change in firing rate between the laser and baseline condition for each stimulus (right). Dots represent recorded units. Bar plots represent means over the population of n = 155 units. Error bars are standard error of the mean. (**F**) Responses to the standard (left), cascade (middle left), and deviant (middle right) for nonadapting units in shell regions of IC under baseline and laser conditions. Change in firing rate between the laser and baseline condition for each stimulus (right). Dots represent recorded units. Bar plots represent means over the population of n = 243 units. Error bars are standard error of the mean.

population exhibiting significant prediction error and repetition enhancement (*Figure 6B*, maroon dots). Some shell adapting units also exhibited a combination of both repetition suppression and prediction error (*Figure 6B*, left). These results suggest that the units in IC exhibit distinct combinations of repetition suppression/enhancement and prediction error/negative prediction error.

## Facilitating units exhibit true repetition enhancement

Facilitating units in both central and shell regions of IC exhibited repetition enhancement at baseline, as defined by the difference in firing rate to the last standard and the same tone embedded in the cascade sequence (*Figure 3G and H*). We sought to further characterize the response to the standard

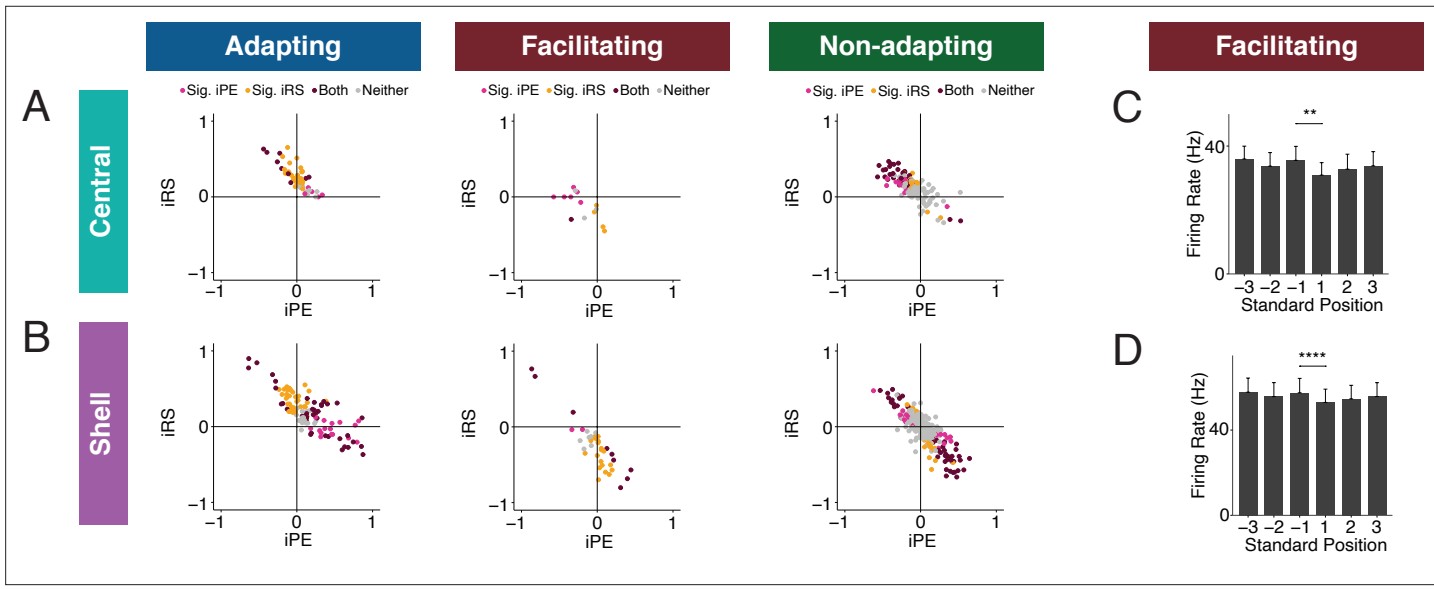

**Figure 6.** Inferior colliculus (IC) units exhibit distinct combinations of index of prediction error (iPE) and index of repetition suppression (iRS). (**A**) Distribution of both iRS and iPE in adapting (left), facilitating (middle), and nonadapting (right) units in central IC. (**B**) Plots of distributions of both iRS and iPE in adapting (left), facilitating (middle), and nonadapting (right) units in shell IC. (**C**) Response to three subsequent standards prior to or following the deviant in facilitating units in central IC. Comparison between the last standard before and the first standard after the deviant demonstrates significant repetition enhancement. Bar plots represent means over the population of n = 14 units. Error bars are standard error of the mean. (**D**) Response to three subsequent standards prior to or following the deviant in facilitating units in shell IC. Comparison between the last standard before and the first standard after the deviant demonstrates significant repetition enhancement. Bar plots represent means over the population of n = 38 units. Error bars are standard error of the mean.

context to determine whether the repetition enhancement captured by the iRS indicates true repetition enhancement (an incremental increase in firing rate on subsequent presentations of the standard) or simply a net increase in firing rate to the standard versus cascade condition. We calculated the mean firing rate for each of the three standards before the deviant and each of the three standards after the deviant (*Figure 6C and D*). The progression of standards by position exhibited subsequent enhancements in firing rate that was plateaued by the second to last standard before the deviant for both central (*Figure 6C*) and shell facilitating units (*Figure 6D*). The firing rate to the last standard was significantly higher than the first in both regions (*Figure 6C*, *Table 1*; p=0.0017, Wilcoxon signed-rank test; *Figure 6D*, *Table 1*; p=9.3e-05, Wilcoxon signed-rank test). These data provide evidence that facilitating units in IC exhibit true repetition enhancement.

## Discussion

### Summary of findings

The results of this study indicate that AC is critically involved in regulating both repetition and prediction effects in the awake IC, providing evidence for the implementation of predictive coding in cortico-subcortical networks. Adapting and facilitating units were bidirectionally modulated by cortico-collicular inactivation, with adapting units becoming less adapting and facilitating units becoming less facilitating on laser trials (*Figure 3*). The decrease in adaptation for adapting units was driven by a decrease in prediction error for units in both central and shell regions of IC ( *Figure 3D*, *Figure 5E*, *Figure 7*, pink arrows). For facilitating and nonadapting units in the central nucleus, inactivation-driven changes were caused by a decrease in repetition enhancement (*Figure 3G*, *Figure 7*, gold dashed arrows). The decrease in facilitation in the shell IC, however, was caused by the abolishment of negative prediction error (*Figure 3H*, *Figure 7*, pink dashed arrows).

In adapting units, these changes were modulated by an increased response to the standard and decreased response to the deviant, while the opposite pattern was true for facilitating units (*Figure 5*). Overall, these bidirectional changes indicate that, without input from AC, IC responds more similarly to tones in the standard and deviant contexts. These findings demonstrate that AC provides critical contextual cues about the statistics of the auditory environment to targets in IC under normal conditions. We further discuss these results in the context of a hierarchical predictive coding framework below.

### iMM in the awake versus anesthetized IC

Our results include the first investigation of how the repetition and prediction processes that underlie deviance detection in the awake IC compare to the anesthetized condition. Our data suggest that while iMM values are higher under anesthesia, they almost entirely reflect repetition suppression, with only a small contribution of prediction error (*Figure 2*). In the central IC, modest prediction error is present under anesthesia, but negative prediction error becomes dominant when the animal is awake. In the shell IC, the same units exhibit drastically different iPE and iRS values for the awake versus the anesthetized condition. Prediction error is substantially higher in the awake IC, and repetition enhancement, rather than repetition suppression, is observed (*Figures 2F and 4G*). These findings suggest that the iMM values in the awake and anesthetized brain reflect different underlying processes, and that anesthesia induces bidirectional changes in metrics of repetition and prediction.

### Facilitating units in IC

We also provide here the first analysis of facilitating units in IC. Previous studies that have investigated iMM have focused selectively on the positive side of the iMM distribution since these units display adaptation. However, facilitation seems to be enriched in the awake IC (*Figures 2B and 4E*) and reflects other potentially interesting parameters, such as repetition enhancement (represented as a higher response to the standard than the cascade sequence) (*Figure 2G*) and negative prediction error (represented as a higher response to the cascade than the deviant) (*Figure 2C*).

### Repetition enhancement and repetition suppression in IC

Because previous studies that have applied a predictive coding framework to decompose neuronal mismatch have focused exclusively on adapting neurons, the repetition enhancement found here

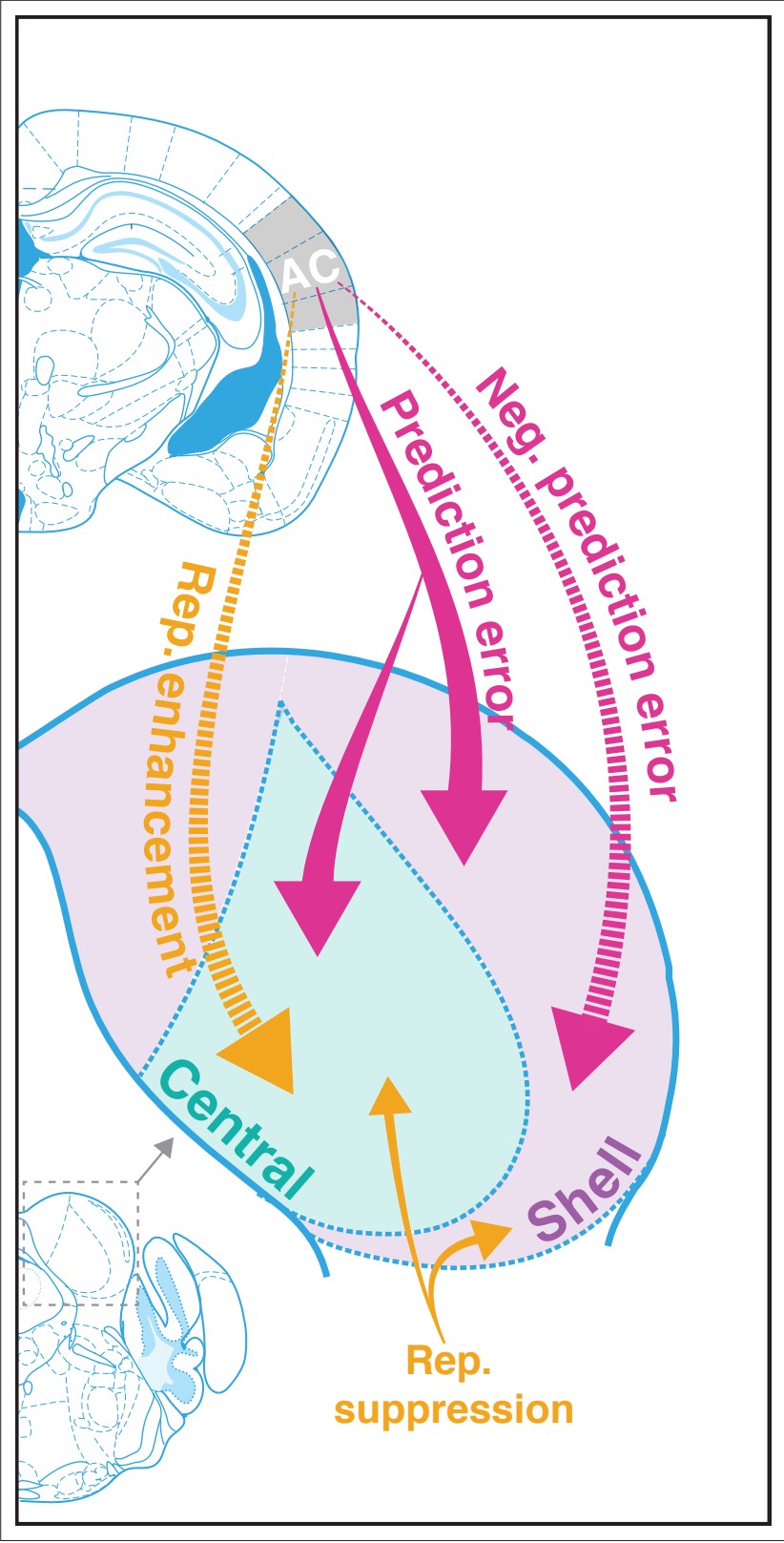

**Figure 7.** Corticofugal regulation of predictive coding. Laser inactivation led to the abolishment of repetition enhancement in central facilitating units and the abolishment of negative prediction error in shell facilitating units. Prediction error decreased during inactivation for adapting units in both shell and central regions of the inferior colliculus (IC). Repetition suppression remained unaffected during cortical inactivation, suggesting that it may reflect fatigue of bottom-up sensory inputs.

in facilitating units has not been previously described (*Parras et al., 2017*). However, it is well-documented in fMRI literature that repetition enhancement is a common phenomenon in humans, existing either alongside or in place of repetition suppression (*de Gardelle et al., 2013*; *Müller et al., 2013*; *Segaert et al., 2013*). Interestingly, repetition enhancement has been proposed to reflect novel network formation and consolidation of novel sensory representations (*Segaert et al., 2013*). Once new representations have been formed, repetition suppression is hypothesized to take over, reflecting the minimization in prediction errors that occurs when new representations give rise to accurate predictions (*Auksztulewicz and Friston, 2016*; *de Gardelle et al., 2013*; *Friston and Kiebel, 2009*). Though the repetition enhancement described in human studies differs drastically on spatial and temporal scales from the phenomenon described here, we find that it similarly involves a sequential enhancement in the response to subsequent presentations of the standard (*Figure 6C and D*). Repetition enhancement has also been observed in the MGB in response to temporally degraded stimuli that are hypothesized to engage top-down resources to compensate for bottom-up acoustic information loss (*Cai et al., 2016*; *Kommajosyula et al., 2019*). Interestingly, this enhancement is reversed when cortico-thalamic pathways are blocked, further suggesting that repetition enhancement in the auditory system reflects a top-down phenomenon (*Kommajosyula et al., 2021*).

While repetition suppression can be understood from a predictive coding framework, it can also be viewed from the perspective of neuronal fatigue, whereby the incremental decrease in firing rate to a repeated standard tone is simply explained by synaptic depression (*Escera and Malmierca, 2014*; *Taaseh et al., 2011*). Interestingly, we did not find any effect on repetition suppression during cortico-collicular inactivation, suggesting that it may reflect fatigue of bottom-up sensory inputs rather than an active predictive process (*Figures 3D and 5E*, *Figure 7*, gold arrows). While these data do not provide definitive proof of either perspective, they do suggest that the processes that underlie repetition suppression in IC do not involve top-down cortical signals. This notion is supported by the fact that repetition suppression was much more prevalent when animals were under anesthesia, a state in which the auditory responsiveness in the cortex is compromised (*Figure 2G*; *Brugge and Merzenich, 1973*; *Katsuki et al., 1959*).

## Prediction error in IC

In both central and shell populations that exhibited prediction error at baseline, cortico-collicular inactivation led to a decrease, or complete abolishment, of prediction error (*Figures 3D and 5E*). According to models of hierarchical predictive coding, higher-order stations generate predictions that they broadcast to lower centers (*Friston and Kiebel, 2009*). These predictions are compared with representations of the actual sensory input, and if there is a mismatch, a prediction error is generated and forwarded up the hierarchy (*Friston and Kiebel, 2009*). Under this framework, the inactivation of top-down inputs would interfere with communication of predictions, leading to dysfunction in the prediction error response, as seen in our data. Another possibility is that prediction errors are directly backpropagated from AC to IC. While this contradicts canonical predictive coding models, evidence for prediction error has been found in deep layers of the cortex in which feedback neurons reside (*Asilador and Llano, 2020*; *Rummell et al., 2016*). Though the precise mechanism underlying the generation of prediction error in IC remains unclear, our data show that feedback from AC plays a critical role in this process.

## Negative prediction error in IC

In addition to units with prediction error, we found that units in IC that responded more strongly to the cascade than the deviant context (*Figure 3G and H*), consistent with previous reports (*Parras et al., 2017*). A stronger response to a tone in the cascade sequence compared to the context in which it is a deviant could simply reflect a relative lack of cross-frequency adaptation; the oddball stimulus consists of repeated tone presentations of two neighboring frequencies, making it more likely to generate cross-frequency effects than the cascade stimulus, which cycles through repetitions of 10 evenly spaced frequencies (*Parras et al., 2017*; *Taaseh et al., 2011*). Previous studies that have investigated the effective bandwidth for cross-frequency adaptation, however, have found that it occurs between channels with a frequency separation of a third of an octave or less (*Taaseh et al., 2011*). The stimuli used in this study had a half-octave frequency separation, indicating that cross-frequency

effects should be minimized. Therefore, it is unlikely that the negative prediction error responses observed in this study simply reflect cross-frequency adaptation to the oddball stimulus.

A stronger response to a tone when it is embedded in a completely predictable sequence, such as the cascade sequence, than when it is a deviant could also signify that a neuron encodes predictions, rather than prediction errors. In hierarchical predictive coding, both predictions and prediction errors are generated at every level of the hierarchy, with prediction errors being forwarded to ascending sensory centers and predictions being backpropagated (*Friston and Kiebel, 2009*). In the shell IC, the region that receives the vast majority of descending cortical input, evidence for negative prediction error was abolished during cortico-collicular inactivation (*Figure 3H*), consistent with the notion that feedback from the cortex may carry predictions to IC (*Bajo et al., 2007*; *Herbert et al., 1991*; *Saldaña et al., 1996*; *Stebbings et al., 2014*). Interestingly, negative prediction error in the central nucleus remained unperturbed during inactivation of cortical feedback (*Figure 3G*). Given that only a small fraction of cortico-collicular fibers terminate in the central nucleus, it is likely that it receives predictions from another source (*Bajo et al., 2007*; *Herbert et al., 1991*; *Saldaña et al., 1996*; *Stebbings et al., 2014*). An intriguing potential candidate for this source of predictions could be the shell IC, given the extensive network of intracollicular connections in IC (*Lesicko et al., 2020*; *Saldaña and Merchán, 1992*; *Saldaña and Merchán, 2005*). Future studies will be required to determine whether the negative prediction error metric described here captures the type of top-down predictions described in canonical predictive coding models.

## Technical considerations

One limitation of this study is that laser inactivation achieved only partial and not complete inactivation of the cortico-collicular pathway. Given that light itself can have a modulatory or toxic effect on neurons, these types of optogenetic experiments require a careful titration between using enough power to substantially affect the population of interest without causing nonspecific light or heat-based perturbations (*Tyssowski and Gray, 2019*). Though other techniques, such as chemogenetic approaches or cooling, provide more complete inactivation, they do not allow for rapid and reversible inactivation (*English and Roth, 2015*). With our laser power parameters, we found a mean 60% reduction in firing in putative cortico-collicular neurons at baseline and a 45% reduction during presentation of pure tone stimuli (*Figure 1—figure supplement 1D*). This reduction produced clear effects on repetition and prediction processing in IC, in several cases with the severe reduction or complete abolishment of certain metrics of deviance detection, such as prediction error and repetition enhancement in the central nucleus and negative prediction error in the shell IC (*Figure 3*). The interpretation of these results should bear in mind that they reflect only partial and not complete inactivation.

The analyses in this study were performed on pooled single- and multiunit data. Although we observed no differences in the iMM distribution between single- and multiunits (*Figure 3—figure supplement 3*), the results of this study should be interpreted with this limitation in mind, namely, photosuppression-induced changes in these units may not reflect changes in single neurons.

Whereas this study focuses on changes specific to the cortico-collicular pathway, it should be noted that cortico-collicular neurons are known to branch to additional subcortical targets besides the IC, including the MGB, caudal regions of the dorsal striatum, and the lateral amygdala (*Asokan et al., 2018*). The fact that our photo-suppression experiments produce short-latency effects in the IC (*Figure 3C and F*) indicates that the observed changes are likely due to direct, monosynaptic AC to IC pathways, and that multisynaptic effects from other collateral sites are unlikely. Nevertheless, the potential contribution from these additional downstream targets cannot be definitely ruled out and should be factored into the interpretation of the results.

## Conclusions

Our findings indicate that deviance detection and predictive coding in IC involve additional complexity than has been previously described. We provide here the first description of facilitating units in IC, as well as evidence for the existence of repetition enhancement and negative prediction error in these units. We show that AC regulates these metrics and is also involved in the generation of prediction error in IC. Repetition suppression is unaffected by inactivation of cortical input to IC, providing evidence that this process may reflect bottom-up fatigue rather than top-down predictive processing.

These results demonstrate the role of AC in providing contextual cues about the auditory stream to targets in IC.

# Materials and methods

## Key resources table

| Reagent type (species) or resource | Designation | Source or reference | Identifiers | Additional information |
|---|---|---|---|---|
| Strain, strain background (*Mus musculus*) | *Cdh23* mice | Jackson Laboratories | *Cdh23tm2.1Kjn*/J; RRID:IMSR_JAX:018399 | |
| Recombinant DNA reagent | AAV9-CAG-FLEX-ArchT-tdTomato | UNC Vector Core | Addgene_28305 | |
| Recombinant DNA reagent | RetroAAV2 hSyn Cre-GFP | In-house | | Vector generated and maintained in the di Biasi lab |
| Software, algorithm | Kilosort2 | Marius Pachitariu | https://github.com/MouseLand/Kilosort; RRID:SCR_016422 | |
| Software, algorithm | MATLAB | MathWorks | https://www.mathworks.com/; RRID:SCR_001622 | |
| Software, algorithm | ImageJ | NIH | RRID:SCR_003070 | |

## Animals

We performed experiments in six adult *Cdh23* mice (*Cdh23tm2.1Kjn*/J, RRID:IMSR_JAX:018399; four males and two females, age 3–8 months). This mouse line has a targeted point reversion in the *Cdh23* gene that protects against the age-related hearing loss common to C57BL/6 strains (*Johnson et al., 2017*). Animals were housed on a reversed 12 hr light–dark cycle with water and food available ad libitum. All procedures were approved by the University of Pennsylvania IACUC (protocol number 803266) and the AALAC Guide on Animal Research. We made every attempt to minimize the number of animals used and reduce pain or discomfort.

## Virus injection

Mice were continuously anesthetized with isoflurane and mounted in a stereotaxic frame. Buprenex (0.1 mg/kg), meloxicam (5 mg/kg), and bupivicane (2 mg/kg) were injected subcutaneously for preoperative analgesia. We performed small craniotomies bilaterally over AC (−2.6 mm caudal to bregma, ±4.3 mm lateral, +1 mm ventral) and IC (−4.96 mm caudal to bregma, ±0.5 mm lateral, +0.5 mm ventral and −4.96 mm caudal to bregma, ±1.25 mm lateral, +1.0 mm ventral). A glass syringe (30–50 µm diameter) connected to a pump (Pump 11 Elite, Harvard Apparatus) was used to inject modified viral vectors (AAV9-CAG-FLEX-ArchT-tdTomato or AAV9-CAG-FLEX-tdTomato; 750 nL/site; UNC Vector Core) into AC and a retroAAV construct (retro AAV-hSyn-Cre-GFP; 250 nL/site) into IC (*Figures 1A and 2A*, *Figure 3—figure supplement 1A*). Large viral injections were performed to broadly target cortico-collicular neurons throughout all regions of the AC. We implanted fiber-optic cannulas (Thorlabs, Ø200 µm Core, 0.22 NA) bilaterally over AC injection sites (0.4 mm ventral to brain surface) and secured them in place with dental cement (C and B Metabond) and acrylic (Lang Dental). IC injection sites were covered with a removable silicone plug (Kwik-Sil). A custom-built headplate was secured to the skull at the midline and a ground-pin was lowered into a small craniotomy over bregma. We injected an antibiotic (5 mg/kg Baytril) subcutaneously for 4 days postoperatively. Virus injection sites were confirmed postmortem for all animals included in the study.

## Extracellular recordings

We performed recordings a minimum of 21 days after virus injection surgeries to allow adequate travel time for the viral constructs (*Figure 1A*). Recordings were carried out inside a double-walled acoustic isolation booth (Industrial Acoustics) or a custom-built table-mounted acoustic isolation booth. For IC recordings, mice were briefly anesthetized to remove the silicone plug over IC virus injection sites. Following recovery from anesthesia, the headplate was clamped within a custom base to provide head-fixation. We lowered a 32-channel silicon probe (Neuronexus) vertically into IC during presentation of broadband noise clicks and monitored sound responses online to confirm localization within IC (*Figure 1A*). In a subset of animals (seven recording sites in two mice), the probe was first coated

in a lipophilic dye (DiD or DiA; Invitrogen) to aid in post hoc reconstruction of recording sites. In each animal, two recordings were performed per IC (four total recording sessions bilaterally). We attempted to target both shell and central IC regions in each animal, and our post hoc analysis of recording sites (see details in 'Analysis' section) revealed that all but one animal was recorded from in both regions. Recordings that did not show significant sound responsiveness were removed from the analysis. Following completion of all IC recording sessions, we recorded the activity of neurons in AC using the same procedure (*Figure 1—figure supplement 1B*). We performed a square craniotomy (2 mm × 2 mm) over AC and oriented the probe vertically to the cortical surface (35° angle of the stereotaxic arm). Electrophysiological data were filtered between 600 and 6000 Hz to isolate spike responses and then digitized at 32 kHz and stored for offline analysis (Neuralynx). For a subset of recordings, the experimental procedures were repeated while recording from the same units after the animal had been anesthetized with isoflurane (*Figure 2A*). We performed spike sorting using Kilosort2 software (https://github.com/MouseLand/Kilosort; RRID:SCR_016422, version 2). Both single and multiunits were included for all analyses (experimental IC: 50 single units, 354 multiunits; control IC: 17 single units; 111 multiunits; anesthetized: 10 single units, 129 multiunits; AC: 95 single units, 300 multiunits; putative cortico-collicular: 9 single units; 11 multiunits).

## Laser inactivation

We inactivated cortico-collicular neurons using a 532 nm DPSS laser (GL532T3-300, Slocs lasers, 3 mW power at cannula tip or OptoEngine, MGL-III-532, 15 mW power at cannula tip) connected via optical fibers to the implanted cannulas (*Figures 1A, 2C and D*). Data collected using either laser was pooled together as no significant differences were observed in the strength of inactivation in AC during silence (p=0.054, Wilcoxon rank-sum test) or the presentation of pure tone stimuli (p=0.072, Wilcoxon rank-sum test) between the two lasers. Square laser pulses were timed to coincide with tone onset and lasted for 100 ms. Evidence of inactivation in putative cortico-collicular units (infragranular AC units with a minimum 30% reduction in both baseline and sound-evoked neuronal activity) was confirmed for all animals included in the study.

## Stimuli

We generated an initial frequency response function from a sequence of 50 pure tones, 1–70 kHz, repeated 20 times at 70 dB SPL in pseudo-random order. This response function was generated online to select suitable frequencies for the oddball stimuli, that is, frequencies that would fall into the average response area for units in a given recording. Each tone was 50 ms duration (1 ms cosine squared ramps) with an inter-stimulus interval of 200 ms and presentation rate of 4 Hz. A similar tuning curve stimulus, with eight amplitude levels (35–70 dB, 5 dB increments) and five repetitions, was used to further characterize the tuning properties of each unit (*Figure 1—figure supplement 2E, F*).

Oddball tone pairs were chosen to fit within the average response area for units from a given recording. Given the prevalence of inhibited regions in the tuning curves, and the fact that this often led to differences in the response profile of the unit to each frequency in the oddball tone pair, the responses to each frequency were analyzed separately (*Figure 1—figure supplement 2G*). Oddball stimuli consisted of a frozen sequence of two pure tones (with the same tone parameters as those used in the initial frequency response functions) with a 90:10 standard-to-deviant ratio and half-octave frequency separation. The number of standards interleaved between two deviants was counterbalanced and varied between 3 and 17 standards. The stimuli were divided into blocks (with the end of a block defined by the presentation of a deviant), and tone type and laser pairings were alternated on subsequent blocks. For example, on the first block the laser stimulus was paired with the deviant, on the second block it was paired with the last standard, and the corresponding tones in the third block served as baseline controls, with no laser stimulus. The number of preceding standards in the blocks was balanced for all three laser conditions (deviant, last standard, and baseline). Each block type (laser + standard, laser + deviant, no laser) was presented 45 times, and the total number of tones in each sequence was 1250. Two oddball sequences were created, both with the same frozen pattern, but with the frequencies of the standard and the deviant switched.

Cascade sequences consisted of either an ascending or descending set of 10 evenly log-spaced (half-octave separation) pure tones (same tone parameters as described above) (*Figure 1C*). The two tones used in the oddball sequences were always included as adjacent tones in the cascade

sequences, though their position within the cascade was varied. To generate the many standards control sequence, we shuffled the cascade sequences using an algorithm that does not allow for repetition of tones of the same frequency on subsequent presentations.

## Analysis

To distinguish between shell and central IC recording locations, we plotted the best frequency for each unit from a given recording against its depth and fit the data with a robust linear regression model (*Figure 1—figure supplement 2B*). Additionally, we computed the mean sparseness for all units from a given recording site to quantify the sharpness of tuning. The $R^2$ metric from the linear fit and the mean sparseness from each recording were used to perform k-means clustering with two groups. Each recording was assigned to a location (either central or shell) according to the k-means output, with central sites typically having high sparseness and high $R^2$ values and shell sites having low sparseness and low $R^2$ metrics (*Figure 1—figure supplement 2C*).

Sound response profiles were categorized quantitatively from analysis of the combined responses to the standard and deviant tones using MATLAB's 'findpeaks' function with a minimum peak height set to the mean of the baseline period (50 ms before tone onset) ± 3 SDs. Units that did not display maxima or minima during the tone duration period (0–50 ms) or in the 50 ms after (the 'offset window') were labeled as sound unresponsive and were removed from the analysis. Units that showed only a single minimum ('inhibited' units) or only a response in the offset window were similarly removed from the analysis. Units that showed at least one maxima during the tone duration period were included in the analysis and further categorized as either onset (single maxima in the first 10 ms after tone onset), sustained (single maximum after the first 10 ms after tone onset), E-I or I-E (units that displayed both a maximum and minimum during the tone duration period), biphasic (units that displayed two maxima during the tone duration period), or mixed (units with greater than two maxima and/or minima during the tone response period). It was common for units to display a response both during the tone duration window and the offset window, and in these cases a combined response profile was assigned (e.g., onset/offset, sustained/inhibited offset). Units with only inhibited or offset responses were removed from the dataset.

Significant adaptation or facilitation for each unit was assessed with a Wilcoxon rank-sum test between the trial-by-trial firing rates to the standard and deviant on the 45 baseline trials. The iMM, identical to the traditional SSA index, was further deconstructed into an iPE and an iRS such that iMM = iPE + iRS. The raw firing rates to the standard, cascade, and deviant conditions were normalized by dividing by the Euclidean norm, $N = \sqrt{FR_{Dev}^2 + FR_{Casc}^2 + FR_{Stan}^2}$ . The iPE was calculated as the difference in normalized firing rate to the deviant and cascade conditions (iPE = $\frac{FR_{Dev}}{N} - \frac{FR_{Casc}}{N}$), while the iRS was calculated as the difference in normalized firing rate to the cascade and standard conditions (iRS = $\frac{FR_{Casc}}{N} - \frac{FR_{Stan}}{N}$). Predictive coding metrics for the laser condition were calculated similarly, but using trials from laser + standard, laser + cascade, and laser + deviant pairings.

## Statistical analysis

Shapiro–Wilk tests were used to assess normality. For normally distributed data, Student's *t*-tests were performed. When the assumption of normality was violated, Wilcoxon rank-sum tests were used for nonpaired data and Wilcoxon signed-rank tests were used for paired data. Cohen's d was calculated as a measure of effect size for *t*-tests. For Wilcoxon tests, the effect size r was calculated as the z statistic divided by the square root of the sample size.

## Acknowledgements

We thank the members of the Geffen laboratory for helpful discussions. This work was supported by funding from the National Institute of Health grants F32MH120890 to AMHL, R01DC015527, R01DC014479 and R01NS113241 to MNG, F31DC016524 to CA, and R01DA044205, R01DA049545 and U01AA025931 to MDB.

## Additional information

### Funding

| Funder | Grant reference number | Author |
|---|---|---|
| National Institute on Deafness and Other Communication Disorders | F32MH120890 | Alexandria MH Lesicko |
| National Institute on Deafness and Other Communication Disorders | R01DC015527 | Maria N Geffen |
| National Institute on Deafness and Other Communication Disorders | R01DC014479 | Maria N Geffen |
| National Institute of Neurological Disorders and Stroke | R01NS113241 | Maria N Geffen |
| National Institute on Deafness and Other Communication Disorders | F31DC016524 | Christopher F Angeloni |

The funders had no role in study design, data collection and interpretation, or the decision to submit the work for publication.

### Author contributions

Alexandria MH Lesicko, Conceptualization, Data curation, Formal analysis, Funding acquisition, Investigation, Methodology, Software, Validation, Visualization, Writing – original draft, Writing – review and editing; Christopher F Angeloni, Data curation, Investigation, Resources, Software; Jennifer M Blackwell, Data curation, Software; Mariella De Biasi, Resources, Validation; Maria N Geffen, Conceptualization, Data curation, Formal analysis, Funding acquisition, Investigation, Methodology, Project administration, Resources, Supervision, Validation, Visualization, Writing – original draft, Writing – review and editing

### Author ORCIDs

Alexandria MH Lesicko http://orcid.org/0000-0003-4766-2258
Maria N Geffen http://orcid.org/0000-0003-3022-2993

### Ethics

Animals were housed on a reversed 12-hour light-dark cycle with water and food available ad libitum. All procedures were approved by the University of Pennsylvania IACUC (protocol number 803266) and the AALAC Guide on Animal Research. We made every attempt to minimize the number of animals used and to reduce pain or discomfort.

### Decision letter and Author response

Decision letter https://doi.org/10.7554/eLife.73289.sa1
Author response https://doi.org/10.7554/eLife.73289.sa2

## Additional files

### Supplementary files

• Transparent reporting form

### Data availability

The data is available for review on the dryad depository, https://doi.org/10.5061/dryad.m905qfv13.

The following dataset was generated:

| Author(s) | Year | Dataset title | Dataset URL | Database and Identifier |
|---|---|---|---|---|
| Geffen MN, Lesicko A, Angeloni C, Blackwell J, De Biasi M | 2021 | Data from: Cortico-Fugal Regulation of Predictive Coding | https://dx.doi.org/10.5061/dryad.m905qfv13 | Dryad Digital Repository, 10.5061/dryad.m905qfv13 |

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
