## [Editor Report]

This study concerns the neural representation of prediction in the central auditory pathway. The authors report that top-down inputs from the auditory cortex carry contextual cues that enable subcortical neurons to distinguish between predictable and unexpected sounds. This work provides important insights into how feedback pathways in the auditory system modulate feedforward signals in a context-dependent fashion.

---

## [Decision Letter]

**Decision letter after peer review:**

[Editors’ note: the authors submitted for reconsideration following the decision after peer review. What follows is the decision letter after the first round of review.]

Thank you for submitting the paper "Cortico-fugal regulation of predictive coding" for consideration at *eLife*. Your submission has been reviewed by three peer reviewers, including Jennifer Groh as the Reviewing Editor and Reviewer #3, and the evaluation has been overseen by a Senior Editor. Although the work is of interest, we are not convinced that the findings presented have the potential significance that we require for publication in *eLife*.

Specifically, after discussion among the reviewers, the most important consensus concerns that emerged were (a) whether the findings are novel in comparison to similar existing studies, including those that involved cooling in auditory cortex and the impact of such cooling on the IC, and (b) the suggestion that some of the reported effects could be due to regression to the mean. This is a potentially addressable problem via the suggestion of Reviewer 2 point 2b. Finally (c) it was noted that the central/shell distinction is critical to the novelty of the findings. Histological confirmation of the assignment of sites to these subdivisions would strengthen the paper. While there were some differences of opinion regarding the clarity of the manuscript, we hope you find even the more critical comments useful.

*Reviewer #1:*

In this study Lesicko and colleagues have studied the effect of AC inactivation using the optogenetic technique to analyze neuronal mismatch in the interior colliculus of the awake mouse.

The study is interesting and is potentially beneficial for the people working on predictive coding. However, the manuscript is long and could be substantially trimmed and focused to make it more useful. While I was originally excited about the manuscript, my enthusiasm decreased after I read it. I had to read several times to make sense and get a general idea of results and still I am not totally convinced about the data and the presentation. In its present form I cannot recommend acceptance as in my humbly view this manuscript needs to be substantially revised.

As opposed to what the authors claims, and after close inspection of previous studies by Parras´ and by their team, it seems to me much of the basic/general results are similar to Parras and colleagues or even Duque and Malmierca. These authors also studied SSA and iMM in awake mouse. The main results are similar. Low levels of iMM and iPE in the IC. This is a major issue for me. Given the low levels of SSA/iMM, I wonder how authors consider if a neuron shows significant iMM…(bootstrapping??) as AC may have mostly subtle changes on the IC responses.

Another major issue is that authors claim that have recorded separately neurons from the central nucleus and the shell. However, they don´t show any histological probe of the electrode recording. The central nucleus in mouse is very small and although it shows a distinct tonotopic organization and response are areas are usually V shaped, these responses can also occur in the shell. So, my concern is that most of the neurons may be actually recorded in the shell. I would like to draw the attention to the authors that previous studies on subcortical SSA/iMM have demonstrated the lack of adapting responses in the central nucleus. Of course, all previous studies may have missed this, but I dare to suggest that much of the authors central nucleus data may actually be from the shell. In any case, this needs to be unambiguously demonstrated here with some histological data; it is not adequate to rely solely on the frequency response areas, Even if this is shown, I would like to see a convincing conceptual framework to understand it.

Also, the conclusion that the cascade and many-standard controls yield similar results has also been reported in Parras and Casado-Roman recently.

The most original part of this study is the in-depth analysis of the repetition enhancement responses, but I also would like to draw the authors´ attention to the previous study by Parras where they also reported negative iPE (cf. Figures3 rat and Figure 7 awake mouse). This is mentioned on page 29, lines 694-696, but it should be more clearly stated in abstract etc. so that previous studies get a fairer recognition. Also, Duque and Malmierca already have reported that SSA (which reflects iMM) is lower in awake than in anesthetized mouse. This is mostly due to the high rate of spontaneous activity, which is not mentioned in this study.

The most interesting part of the paper is the section related to the effect of cortical deactivation. However, as the authors themselves note, the technique has some limitations. I would like a more detailed discussion on how such limitations may have affected the results and hence the conclusions. Another important issue not addressed is where in AC the injections were made (A1, AAF, A2, etc ???) and how large the injection sites are. These technicalities may have affected the results and should also be considered and different fields may have different projections to IC.

*Reviewer #2:*

Lesicko et al. studied the role of corticofugal feedback in predictive coding in the auditory system, building on previous studies of stimulus-specific adaption. With a focus on the IC, the authors measured contextual effects on tone responses with and without Arch-mediated inactivation of AC neurons that project to the IC. Using a tone cascade stimulus as a baseline, they divided SSA effects into adaptation to a regular, repeating stimulus (repetition suppression) and enhanced responses to an oddball stimulus (prediction error). This work nicely replicates some previous findings, including the relatively low rates of SSA in IC, especially central areas, the decrease in SSA magnitude for awake versus anesthetized animals, and the decrease in SSA magnitude following inactivation of auditory cortex. By breaking SSA effects into its components, they are able to argue that feedback from AC primarily signals prediction error (rather than suppression). In addition, they identify a group of neurons that shows an opposite pattern to standard SSA, with enhanced responses to repetition and decreased responses to oddball stimuli.

Several new observations help refine understanding of the role of cortical feedback in sound processing. Perhaps most substantial is the observation of enhanced repetition responses in the auditory midbrain. While these results may be important, there are some methodological/analytical concerns that should be addressed, especially given that the pattern of repetition enhancement has not been reported previously.

Concerns:

1. The novelty of the current result could be spelled out more clearly. The authors cite previous work from the Malmierca group reporting diminished SSA in awake animals and when cortex was silenced, but don't provide a more detailed comparison. Is the novelty of the current study that the cortical effect was linked specifically to prediction error? But why then is the major effect of Arch on non-adapting cells to increase repetition suppression? While there are some overarching themes, the results currently seem a bit scattered and would be strengthened if linked more directly to the previous work.

2. The report of facilitating neurons is quite interesting, but some important details are not clear.

a. (L. 213-214) It took some close reading, but it appears that neurons with suppressive responses were not analyzed for SSA effects. How were neurons with suppressive responses defined? If a neuron showed a transient response followed by suppression (eg, left panel of Figure 3D), did that count as suppressive? It appears in this example, that a tone actually evokes a sustained response that is suppressive, relative to the spontaneous rate. If repetition suppression is computed from raw firing rate, a decrease in suppression in this case could actually appear as an enhancement. The example in Figure 5F, of course, provides a clear demonstration of facilitation, but it is important to know what the typical response profile is for the adapting versus suppressive neurons and if they are qualitatively different. Perhaps the average PSTH responses could be compared?

b. (L. 386-391) How are neurons defined as adapting vs. facilitating? Based only on non-laser trials? One worries that some of the reported effects may reflect a regression to the mean. That is, if there was experimental noise that made responses slightly adapting or facilitating in the laser-off conditions, then the absence of adaptation or facilitation in the laser-on condition may be that the noise was absent. A potentially less biased approach would be to define adapting versus facilitating neurons based on responses averaged across both laser-on and -off trials. Given that the adaptation effects are relatively infrequent and small, this additional control seems important.

L. 47-48. "suppression … suppression" Not a concern really, but a request. The language is technically correct, but the word suppression refers both to a neural computation (suppression of error signals) and optogenetic manipulation (Arch-mediated suppression). The authors might consider an alternative term for one of these elements of the paper, e.g., "optogenetic inactivation"?

L. 102. "repetition and prediction" unclear. "repetition suppression and prediction error"?

L. 127. What volume of virus was injected in AC and IC?

L. 152. "experimental procedures were repeated" please clarify, were the same units recorded in both conditions?

L. 155. "experimental" does this refer to recordings from IC? Also, please clarify if there were any differences in the results for SUA vs. MUA. This is particularly important for central IC, where single unit isolation is typically difficult.

L. 193. "evenly spaced" Please confirm, "evenly log-spaced"?

L. 217. "iMM, equivalent to…" Does "equivalent to" mean "identical to" the traditional SSA index?

L 218. "iMM = iPE and iRS" Should "and" be "+"?

L 220. "FR" please clarify if spontaneous rate was subtracted or considered as part of the analysis.

L. 262. The term "error suppression" is a bit confusing. Any difference between standard and deviant response (positive or negative) seems like an error signal. Here the term appears to indicate weaker response to the deviant. It's fine if this is a standard term used elsewhere, but the authors might otherwise consider an alternative.

L. 289. What volume of AC was labeled with virus? Was it limited to a tonotopic region? If so, were effects of laser inactivation frequency-specific in IC? This is understandably a difficult question to answer definitively, but some information about the extent of transduction would be helpful.

L. 411. (Figure 5 legend) Were the examples in C and F recorded in central or shell IC?

L. 497. "non-adapting neurons … increase repetition suppression" this result is confusing. Is the idea that two factors cancel each other out, and the optogenetic inactivation unmasks an adaptation process? Some help is needed for interpretation.

L. 681. "According to hierarchical predictive coding" Confusing, maybe "According to models of hierarchical…"?

L. 707. "… stronger response … in a completely predictable sequence" this is an interesting point. If this is the case, should the response to a tone in a cascade differ from the many standards condition?

*Reviewer 3:*

This is an excellent study testing the influence of auditory cortical connections to the inferior colliculus on context-dependent aspects of response patterns in the IC. The authors deploy a paradigm that permits dissociating two different forms of context-dependence, the predictability of a sound and the overall statistics of the sounds, and they use optogenetic methods to investigate the contributions of auditory cortex to how IC neurons alter their responses in these two different types of contexts – this is the particularly novel finding in the paper as the basics of IC neuron response characteristics as a function of context had been previously explored. The manuscript also provides a direct comparison of awake vs anesthetized mouse results, which is a very nice addition and service to the literature as it helps the community incorporate both awake and anesthetized results together and understand their similarities and differences. The manuscript is very well written and the figures are carefully constructed.

[Editors’ note: further revisions were suggested prior to acceptance, as described below.]

Thank you for resubmitting your work entitled "Cortico-fugal regulation of predictive coding" for further consideration by *eLife*. Your revised article has been evaluated by Barbara Shinn-Cunningham (Senior Editor) and Jennifer Groh (Reviewing Editor, Reviewer 1)

The manuscript has been improved, and the reviewers agreed that this work is important, but there are some remaining issues involving the analyses that need to be addressed as outlined in greater detail below (Reviewers 2 and 3).

*Reviewer #1:*

This is a well designed and well written study to investigate the role of descending inputs from auditory cortex to the inferior colliculus in modulating responses to sound. The behavioral task permits a distinction between various forms contextual interpretation of sound, and distinct roles for descending inputs in these contexts are identified. The work replicates and extends prior work in this area, and is likely to be highly impactful regarding our understanding of how the brain's "backwards" connections govern processing in sensory pathways.

*Reviewer #2:*

This study measures the impact of feedback from auditory cortex (AC) to the auditory midbrain (inferior colliculus, IC) on neural encoding of predicable versus unpredictable sounds. Consistent with previous work, the authors find that inactivation of AC feedback reduces differential adaptation to high- versus low-probability stimuli. By introducing a new stimulus condition to their analysis, they provide evidence that the specific effect of AC feedback is to enhance prediction errors and that other forms of adaptation occur independent of AC.

The authors have done a good job addressing concerns raised during the initial review. In particular they have more clearly contrasted their new results on prediction error with previous corticofugal/SSA work that did not distinguish between adaptation and prediction error. This provides a substantial advance on previous work.

However, some concerns do persist around the validity of statistical methods used.

L. 323-324: "To ensure that the laser-induced changes described above were opsin-mediated, we performed control experiments in two mice with identical manipulations to the experimental group, but in the absence of ArchT…"

It feels like nagging, but the circumstances described in this study--where fairly symmetric tails of a distribution both shift toward zero--are exactly the case where regression to the mean is a concern. The control cited above provides evidence that activation of ArchT has some impact on adaptation, but it is not clear that this single control is adequate to support all the subsequent findings in the manuscript. It would be more convincing if the authors could categorize units based on responses averaged across laser-off and -on conditions. If this is not feasible, then the authors should address the following:

1. Are the experimental groups in Figure 3S1B the same as in Figure 3B? Perhaps some details are missing or there is a labeling issue with the x axis? The distributions in 3S1B appear narrower than in 3B. A narrower distribution might indicate less noise, which would reduce the possibility of regression to the mean. Please clarify if in fact the experimental data should be the same and or if differences in the width of the iMM distributions might impact the validity of the control. Are the fractions of adapting/facilitating/non-adapting neurons similar for the control and experimental groups?

2. While the control experiment/analysis supports conclusions reported in Figure 3, it is not clear that it is adequate to support the conclusions in the subsequent analyses, where the data are further processed (e.g., Figure 4C, IRS<0 only) or different quantities are analyzed (e.g., firing rate in Figure 5). One solution would be to run the same analysis on control data in each case. This option does seem cumbersome, and authors might have a better idea for how to address this concern.

L. 88. "However, it remains unknown whether these modulations in the SSA index with cortical deactivation reflect changes in predictive processing." Minor. This sentence seems out of place, as the relationship between SSA and prediction error is not laid out until the next paragraph.

L. 95. "Prediction error…" It might help to rephrase this sentence to provide a definition of prediction error as a component of SSA, in the same way that the previous sentence defines repetition suppression.

L. 253 "… while an iPE value…" Should this be "… while the mean iPE value"? Also, since the same neurons were recorded in both conditions could a paired test be used here? Students T usually treats the two distributions as independent.

*Reviewer #3:*

This manuscript will be of interest to the broad sensory neuroscience field. Prediction error signals, reflecting a mismatch between expected and actual sensory inputs, have been described across sensory modalities and the contributions of bottom-up and top-down processing are still unknown. This work reveals the contributions of descending cortico-collicular inputs to predictive coding of neurons within the inferior colliculus.

In the present manuscript, Lesicko and colleagues studied the contributions of cortico-collicular neurons to the components underlying stimulus specific adaptation (SSA) of neurons within the inferior colliculus (IC). This is a subject of interest for the broad sensory neuroscience field, as SSA has been described across modalities and the contributions of bottom-up and top-down processing are still unknown. Overall, the experimental design and results are convincing, straight-forward and well presented. Although some of these results simply corroborate previous findings, this study does provide the following conceptual advances from prior work:

1. The contribution of the decomposed processes underlying SSA in subdivisions of the inferior colliculus (IC) has been studied previously in awake mice (Parras et al. 2017) and cortical manipulations have demonstrated the influence of cortical feedback on SSA (Anderson and Malmierca 2013). However, this work combined these approaches to evaluate the contribution of cortico-collicular projections to the distinct SSA processes (repetition suppression and prediction error) in the IC of awake mice.

2. By recording from the same units in the IC of mice in awake and anesthetized states, the authors provide evidence for changes in the repetition and prediction processes underlying SSA across behavioral states. This result is significant, as much of the previous work has been performed in anesthetized rodents.

3. The authors focused on previously ignored facilitating and non-adapting neurons. They discover populations of IC neurons that show repetition enhancement and negative prediction error, which were suppressed during cortico-collicular inactivation. This novel finding has important implications for top-down cortical regulation of predictive coding in IC.

Together, the results from this study will contribute to our understanding of predictive coding in the central auditory system. The strength of this study is the rigorous experiments and comprehensive analysis: the authors examined SSA in neurons from distinct IC subdivisions in both awake and anesthetized mice, assessed the specific effects of corticocollicular projections and analyzed previously-ignored facilitation. Addressing the following concerns will greatly increase the significance of the findings:

1. The authors pool together data from single and multi-unit recordings. Although the inclusion of multi-units might not necessarily affect the described results, this limitation should be clearly stated in the "technical considerations" section of the discussion. For instance, the inclusion of multi-unit recordings may underlie the finding of "mixed" firing types in the IC, as shown in Figure 1-S2C. Importantly, the figure comparing responses between single- and multi-unit responses (provided in the response to the reviewers) should be included as a supplementary figure. Moreover, the author's interpretation of Figure 6 is that 'individual neurons exhibit distinct combinations of iPE and iRS'. Unless these data are only representing single units (which would not be consistent with the total number of single-unit recordings shown in the plot given to the reviewers), Figure 6 (panels A,B) fails to demonstrate that single neurons are exhibiting distinct iPE and iRS. This limitation should be clear in the Results section and included in the 'Technical Considerations' section of the Discussion.

2. The data shown in Figure 1-S2 are not convincing that the recordings were obtained from either the central or shell IC. The authors should provide more histological evidence if the data are available. For example, the panel showing the recording site within the central IC (Figure 1-S2D) also shows some DiA signal in the shell. The authors claim to have performed histological reconstruction by DiD/A probe coating in a subset of animals (lines 708-709). The authors should state how many recording sites used in the study were evaluated histologically, and include all data in a supplemental figure. In addition to the histology, the authors also use the response properties of the IC neurons to distinguish between central and shell recordings. This is a nice complementary method. However, while there are sites clearly distinguished as central or shell recordings based on sparseness and correlations between BF and depth, some sites are borderline (Figure 1-S2C). For example, several sites with high mean sparseness (characteristic of central IC) are categorized as shell recordings based on low correlations between BF vs depth. Could these sites instead be recordings in central IC with an electrode penetration angle slightly off the tonotopic axis? The differences between central and shell IC neurons in predictive coding, effects of anesthesia, and effects of cortico-collicular silencing are interesting findings. If the authors could provide additional data to give us more confidence in their ability to distinguish these sites, these results would be more compelling.

3. Figure 1-S1C shows an off/rebound response for cortico-collicular neurons when the laser is turned off. The iMM (index of neuronal mismatch) is calculated by comparing the firing in response to a standard and a deviant tone that occurs 200ms after the standard. Thus, rebound activation of the cortico-collicular neurons after the last standard stimulus may alter the response to the subsequent deviant stimulus, impacting iMM. This potential confound should be discussed within the 'technical limitations' section of the Discussion.

4. Figure 1-S1A. Although the study emphasizes the specific manipulation of the cortico-collicular projections, it should be noted that auditory corticofugal neurons that innervate the IC have other widespread targets including the medial geniculate body, striatum and lateral amygdala (Asokan et al., 2018, Nature Communications). Did the authors also see expression of axons in widespread brain regions with their viral strategy? If so, this could be shown in this panel or a supplemental figure. The involvement of these other regions in the inactivation studies should be addressed in the Discussion.

---

## [Author Response]

[Editors’ note: the authors resubmitted a revised version of the paper for consideration. What follows is the authors’ response to the first round of review.]

Reviewer #1:1. “the manuscript is long and could be substantially trimmed and focused to make it more useful”.

We have revised the present draft of the manuscript to make it more focused. Specifically, we have moved Figure 2 and 3 to the supplementary material, as these figures are meant to provide validation of the experimental methods and not new findings.

2. “it seems to me much of the basic/general results are similar to Parras and colleagues or even Duque and Malmierca. These authors also studied SSA and iMM in awake mouse. The main results are similar. Low levels of iMM and iPE in the IC. This is a major issue for me.”

We have extensively studied Parras et al., 2017 and Duque and Malmierca, 2015 as they provide much of the groundwork and inspiration for the present study (Duque and Malmierca, 2015; Parras et al., 2017). We have emphasized in the abstract and discussion several novel findings. Specifically:

– Like Parras et al., 2017, we decomposed stimulus specific adaptation into two distinct processes: prediction error and repetition suppression. A novel contribution is that we specifically determined how corticofugal inputs affect prediction error and other metrics of deviance detection in the inferior colliculus.

– While we identify “low levels of iMM and iPE in the IC”, it is not the main result of the study. Our main findings are that top-down inputs from the auditory cortex regulate prediction error, as well as other metrics such as repetition enhancement, in the IC, and that the cortex routes contextual information subcortically.

– As mentioned by reviewer #2, Anderson et al. 2013 previously investigated how cortico-collicular inputs affect SSA in the inferior colliculus through cortical deactivation in anesthetized rats (Anderson and Malmierca, 2013). In the present study we also deactivate cortico-collicular inputs and measure SSA. However, we furthermore determine how the index of prediction error, and the index of repetition suppression are affected by deactivation, not just SSA as a whole. This distinction is critical, as these two metrics reflect different underlying processes, whereas previously reported results could have captured changes in either and/or both. It is also worth noting that these findings are derived from awake animals, while the previous study used anesthetized animals. Our study presents novel data that show that the prediction and repetition processes that are reflected in the SSA index differ substantially between the awake and anesthetized condition, with prediction error and repetition enhancement being much more prevalent in the absence of anesthesia. It is likely that previously reported findings reflect different underlying processes than those studied here in the awake animal, further emphasizing the novelty of the current study.

– Duque and Malmierca, 2015 reported whether and how anesthesia and spontaneous activity affect SSA in the inferior colliculus. The authors conclude that SSA is “similar, but not identical, in the awake and anesthetized preparations” and show that the differences are “mostly due to the higher spontaneous activity observed in the awake animals.” We also compare how anesthesia affects SSA, but we (1) record from the same units while the animal is awake and under anesthesia to directly compare the two conditions and (2) further assess how the index of prediction error and the index of repetition suppression are affected. We find that SSA in awake animals reflects entirely different underlying processes than those in anesthetized animals. Specifically, prediction error and repetition enhancement are significantly more prevalent in shell IC units when animals are awake, while low levels of prediction error and high repetition suppression dominate under anesthesia. In the central nucleus, negative prediction error becomes dominate when the animal is awake. Given that much of the prior research on SSA in the auditory midbrain has been conducted in anesthetized animals, we believe this is a finding of major significance for the field and suggests that the state of anesthesia must be taken into consideration when interpreting SSA studies.

3. “Given the low levels of SSA/iMM, I wonder how authors consider if a neuron shows significant iMM…(bootstrapping??) as AC may have mostly subtle changes on the IC responses”.

We performed this analysis. This information is provided in lines 791-792 of the present manuscript: “Significant adaptation or facilitation for each neuron was assessed with a Wilcoxon rank sum test between the trial-by-trial firing rates to the standard and deviant on the 45 baseline trials.”

We want to emphasize that the fact that there are relatively low levels of SSA/iMM (in comparison to the cortex) is not a major point or conclusion of the present study; rather, we are interested in how inputs from the cortex affect the indices of neuronal mismatch, prediction error, and repetition suppression in IC units. A “low level” of SSA/iMM does not necessarily mean that a unit is absent of prediction error, etc. For example, we find that a high level of prediction error and a negative index of repetition suppression (indicating repetition enhancement) can result in a SSA index/iMM close to zero (see Figure 2E-G; Figure 6).

4. “Another major issue is that authors claim that have recorded separately from the central nucleus and the shell. However, they don’t show any histological probe of the electrode recording.”

We have added histological data to the manuscript from a recording that was categorized as a central nucleus site and a recording that was categorized as a shell site (Figure 1 —figure supplement 2D) using the analytic methods detailed in lines 768-775 of the present manuscript. In both instances the recording electrode was submerged in a lipophilic dye prior to tissue insertion to mark the recording location. An atlas image overlay was used to define the locations of the central (denote as “CIC” here) and shell (denoted as “ECIC” and “DCIC” here) regions of the IC (Paxinos and Franklin, 2019). Notably, our histological data provide the same categorization of electrode sites as the analytical methods used in the present manuscript.

5. “The central nucleus in mouse is very small and although it shows a distinct tonotopic organization and response are areas are usually V shaped, these responses can also occur in the shell. So, my concern is that most of the neurons may be actually recorded in the shell.”

Indeed, responses in the shell can be V-shaped and regionally tonotopic (Barnstedt et al., 2015; Wong and Borst, 2019). However, the shell does not exhibit the same stereotyped tonotopic gradient with depth that is highly characteristic of the central nucleus of the IC (see Figure 3B) (Aitkin et al., 1975; Malmierca et al., 2008; Stiebler and Ehret, 1985; Syka et al., 2000). Our method for parsing the recording sites considers the patterns in best frequency and sparsity in all units across the entire depth of the electrode – whereas it is possible to encounter some V-shaped responses or local regions of tonotopy in the shell, it is unlikely that these characteristics will produce the same highly linear fits and consistent high sparsity (reflecting V-shaped tuning) that we find with our central nucleus recordings. Because histological assessment alone can render inconclusive subdivision assignments, specifically in instances when the recording location is near a boundary between subdivisions, these methods allow for an unbiased analysis of site location.

6. “I would like to draw the attention to the authors that previous studies on subcortical SSA/iMM have demonstrated the lack of adapting responses in the central nucleus. Of course, all previous studies may have missed this, but I dare to suggest that much of the authors central nucleus data may actually be from the shell.”

This is an important concern, and we compared our results more directly to those reported in the literature. We found comparable mean baseline SSA/iMM values for the adapting units in the central nucleus (mean = 0.26, Figure 4A and Table 1) to those in the literature for awake mouse (mean = 0.24, Parras et al., 2017 Table 2). We also find that there are fewer adapting units in the central nucleus compared to the shell (Figure 3B,), and those adapting units have lower SSA/iMM indices than their counterparts in the shell (Figure 3B), also in line with previous studies (Duque et al., 2012; Parras et al., 2017). Furthermore, if most of the recordings categorized as central were actually recorded in the shell, we would likely see similar trends in predictive coding metrics from both regions. In fact, we find that both the baseline values and the laser-induced changes in these regions are distinct.

7. “In any case, this needs to be unambiguously demonstrated here with some histological data; it is not adequate to rely solely on the frequency response areas, even if this is shown, I would like to see a convincing conceptual framework to understand it.

We agree and we have added histological data as detailed in response to comment 4. We exploit known differences in the tonotopic organization across the depth of the tissue between shell and central regions to further parse the recording sites. This pattern of increasing best frequency with depth has been very well categorized (see response to point #5) and has been similarly employed by other researchers to categorize IC recordings sites, as mentioned by reviewer #3 (Bulkin and Groh, 2011; Ress and Chandrasekaran, 2013). We have edited the manuscript to include these references and expand the conceptual framework that justifies the use of these analytical techniques for sorting recording sites in lines 202-222 of the present manuscript.

8. “Also the conclusion that the cascade and many-standard controls yield similar results has also been reported in Parras and Casado-Roman recently.”

We thank the reviewer for bringing this study to our attention and have cited it in lines 375-376 of the edited manuscript (Casado-Román et al., 2020). The fact that the cascade and many-standards sequences yield similar results is not a major conclusion of this study (Figure 3 —figure supplement 2), but rather an important control to validate the use of the cascade sequence in further decomposing the iMM into an index of prediction error and an index of repetition suppression. We find it reassuring that Parras and Casado-Roman have also found little difference between the use of these two control sequences.

9. “The most original part of this study is the in-depth analysis of the repetition enhancement responses, but I also would like to draw the authors’ attention to the previous study by Parras where they also reported negative iPE (cf. Figures3 rat and Figure 7 awake mouse). This is mentioned on page 29, lines 694-696, but it should be more clearly stated in abstract etc. so that previous studies get a fairer recognition.

This is a great suggestion. Repetition enhancement reflects a higher response to the standard context than to the cascade context, resulting in a negative iRS value. This is different than the negative iPE value mentioned in the Parras et al. study, which we have termed “negative prediction error” in the present manuscript. We are aware that Parras et al. also found units with negative iPE and have contrasted their interpretation of these results with our own in the Discussion (lines 608-638). Whereas Parras et al. do find units with negative iRS indices, they do not show that negative iRS reflects repetition enhancement (Figure 6C,D), which we report for the first time here. Further, one of our main results is that repetition enhancement decreases during cortical suppression, suggesting that it is a top-down phenomenon, which is a novel finding.

10. “Also, Duque and Malmierca already have reported that SSA (which reflects iMM) is lower in awake than in anesthetized mouse. This is mostly due to the high rate of spontaneous activity, which is not mentioned in this study.”

This is an insightful comment. We also find that SSA index/iMM is lower in awake vs. anesthetized animals (Figure 2). However, this is not a major conclusion of the present study. Rather, we sought to determine what underlying repetition/prediction processes the SSA index/iMM reflects in the awake vs. anesthetized condition, as these processes have drastically different functional implications for predictive processing. We show that prediction error and repetition enhancement are significantly more prevalent in the awake animal in shell IC units, and that negative prediction error dominates the awake central nucleus.

11. “The most interesting part of the paper is the section related to the effect of cortical deactivation. However, as the authors themselves note, the technique has some limitations. I would like a more detailed discussion on how such limitations may have affected the results and hence the conclusions.”

We appreciate the reviewer’s recognition of the merits of the study. We discuss in the manuscript that the main drawback of using laser photosuppression to mediate cortico-collicular deactivation is that it does not achieve full inactivation (lines 641-654). Indeed, this is a concern to us, but we believe the inactivation is robust. We found a mean 60% reduction in firing in putative cortico-collicular neurons at baseline and a 45% reduction during presentation of pure tone stimuli with our laser parameters and observed clear effects on repetition and prediction processing in IC.

12. “Another important issue not addressed is where in AC the injections were made (A1, AAF, A2, etc ???) and how large the injection sites are. These technicalities may have affected the results and should also be considered, and different fields may have different projections to IC.”

We have provided these details in lines 687-692 of the manuscript: “A glass syringe (30-50 µm diameter) connected to a pump (Pump 11 Elite, Harvard Apparatus) was used to inject modified viral vectors (AAV9-CAGFLEX-ArchT-tdTomato or AAV9-CAG-FLEX-tdTomato; 750 nL/site; UNC Vector Core) into AC and a retroAAV construct (retro AAV-hSyn-Cre-GFP; 250 nL/site) into IC (Figure 1A, 2A, Figure 3 —figure supplement 1A). Large viral injections were performed to broadly target cortico-collicular neurons throughout all regions of the auditory cortex.” We agree that it could be informative to consider whether projections from specific sub-fields differentially affect metrics of predictive coding and deviance detection in the IC, and that would be an interesting future direction.

13. “Figures are difficult to read. I am not sure if this is due to the pdf that the system generates, but the dot raters of responses etc. e.g., Figure 2, Figure 5, 6. etc. (dots in scatter plots) are almost impossible to make out and I have to simply rely on the text. The figure quality needs to be significantly improved in order to see the data.”

Thank you for pointing this out. We have increased the dot size and the panel size for the figures mentioned and several others in order to improve the visibility of the data.

Reviewer #2:1. “The novelty of the current result could be spelled out more clearly.”

We have edited the manuscript in several places to address this issue (lines 40-41; 86-90; 103-104; 238244; 404-405) and have included a summary diagram in Figure 7 to further elucidate the main findings. Please see responses to reviewer 1 comments 2, 8, 9 and 10.

We also provide below a summary of the major findings from the present study:

– We show for the first time that cortical input is critical for generating prediction error in IC units, suggesting that the cortex regulates predictive coding subcortically. To our knowledge, this is the first demonstration of predictive coding in a cortico-subcortical network, as virtually all prior studies have focused on cortico-cortical interactions.

– We show for the first time that repetition suppression is unaffected by cortical inactivation, suggesting that this process may reflect fatigue of bottom-up sensory inputs rather than deviance detection.

– We also show for the first time that a subset of IC neurons exhibit repetition enhancement. Further, we show that repetition enhancement is abolished in the absence of cortical input for central IC units, suggesting that it is a top-down phenomenon.

– We show that cortico-collicular inactivation leads to bidirectional changes in the response to the standard vs. deviant tone contexts, such that IC cells respond more similarly to both contexts in the absence of cortical input. These findings suggest that under normal conditions the cortex routes contextual information to the IC.

– We provide the first direct comparison of SSA (i.e., in the same units) in awake vs. anesthetized conditions. We show that SSA reflects drastically different repetition and prediction processes in awake vs. anesthetized animals: in the central IC, negative prediction error rather than prediction error dominates when the animal is awake, and in the shell IC, prediction error and repetition enhancement are significantly more prominent. These findings have important implications for the field, as the vast majority of previous SSA studies have been conducted in anesthetized animals.

2. “The authors cite previous work from the Malmierca group reporting diminished SSA in awake animals and when cortex was silenced, but don’t provide a more detailed comparison.”

Please see response to points 2, 8, 9 and 10 of reviewer #1.

3. “Is the novelty of the current study that the cortical effect was linked specifically to prediction error? But why then is the major effect of Arch on non-adapting cells to increase repetition suppression? While there are some overarching themes, the results currently seem a bit scattered and would be strengthened if linked to the previous work.”

One main finding is that cortico-collicular deactivation decreases prediction error in IC units. However, we also show that the cortex plays a role in repetition processing; specifically, we find that repetition enhancement is abolished in facilitating units in the central IC, suggesting that it may be a top-down phenomenon (Figure 3G). As the reviewer mentions, we additionally find an effect on repetition processing in central non-adapting cells with cortical deactivation, leading to an enhanced index of repetition suppression. This change also reflects a decrease in repetition enhancement. We have added data to better illustrate this finding (Figure 4C, bottom) showing that when the iRS is further parsed for non-adapting units, it is those with negative indices (i.e. those that show repetition enhancement) that show a significant laser effect. We believe that this finding further solidifies the notion that repetition enhancement is a top-down phenomenon.

4. “How were the neurons with suppressive responses defined? If a neuron showed a transient response followed by suppression (eg, left panel of Figure 3D), did that count as suppressive? It appears in this example, that a tone actually evokes a sustained response that is suppressive, relative to the spontaneous rate. If repetition suppression is computed from raw firing rate, a decrease in suppression in this case could actually appear as an enhancement.”

“Suppressed units”, henceforth referred to as “inhibited” units were defined quantitatively from analysis of the combined responses to the standard and deviant tones using MATLAB’s “findpeaks” function with a minimum peak height set to the mean of the baseline period (50 ms before tone onset) +/- 3 standard deviations. In brief, units that displayed a single minimum peak in the PSTH during the 0-50 ms tone duration were termed either “inhibited” units or “inhibited onset” units (if the peak of the minimum occurred in the first 10 ms after tone onset). These units were removed from the analysis. Units that showed a combination of a minimum and maximum during tone duration were labeled as either E-I or I-E (excited-inhibited or inhibited-excited) units, depending on whether the minimum or maximum occurred first. These units were included in the analysis such that their excitatory sound responses could be analyzed (see lines 776-789 for further details of categorizing sound response profiles).

5. “The example in Figure 5F, of course, provides a clear demonstration of facilitation, but it is important to know what the typical response profile is for the adapting versus suppressive neurons and if they are qualitatively different. Perhaps the average PSTH responses could be compared?”

We thank the reviewer for this excellent suggestion and have replaced the single examples with the average PSTH responses for adapting and facilitating units in the revision. The average PSTH for both adapting and facilitating units show excitatory, rather than inhibited responses (Figure 3C,F).

6. “How are the neurons defined as adapting vs. facilitating? Based only on non-laser trials? One worried that some of the reported effects may reflect a regression to the mean. That is, if there was experimental noise that made responses slightly adapting or facilitating in the laser-off conditions, then the absence of adaptation or facilitation in the laser-on condition may be that the noise was absent. A potentially less biased approach would be to define adapting versus facilitating neurons based on responses averaged across both laser-on and -off trials. Given that adaptation effects are relatively infrequent and small, this additional control seems important.”

We edited the methods to state: “Significant adaptation or facilitation for each neuron was assessed with a Wilcoxon rank sum test between the trial-by-trial firing rates to the standard and deviant on the 45 baseline trials.” (lines 791-792).

We agree that it is critical to rule out that the observed effects are simply due to regression to the mean. We took multiple steps at the study design stage to control for regression to the mean, namely using multiple baseline measurements and including a control group to provide an estimate of the change caused by regression to the mean, both standard practices that “can be combined to give even greater protection against regression to the mean” (Barnett et al., 2005). Units were defined as adapting/facilitating/non-adapting based on a statistical comparison of the trial-by-trial responses to the standard and deviant on 45 separate baseline trials. Although it is possible to have experimental noise on a given baseline trial that is not present in a laser trial, the use of multiple baseline trials protects against the possibility that a spurious extreme value will affect the overall categorization of the units as adapting/facilitating/non-adapting. We also included a control group that underwent identical manipulations to our experimental group except ArchT was not present in the viral construct that was injected in the auditory cortex (Figure 3 —figure supplement 1). For this group, we observed no significant differences in iMM, iPE, or iRS for any of the adapting, facilitating, or non-adapting cells in both the central and shell regions. Given that this control group provides an estimate of the change caused by regression to the mean, we conclude that the significant effects in the experimental group were not caused by this phenomenon.

7. L. 47-48. "suppression … suppression" Not a concern really, but a request. The language is technically correct, but the word suppression refers both to a neural computation (suppression of error signals) and optogenetic manipulation (Arch-mediated suppression). The authors might consider an alternative term for one of these elements of the paper, e.g., "optogenetic inactivation"?

Per the reviewer’s suggestion, we have edited the manuscript to avoid using “suppression” to describe different processes:

– Photosuppression/cortical suppression refers to Arch-mediated suppression of cortico-collicular neurons.

This term will be changed to “inactivation” or “optogenetic inactivation” throughout the manuscript.

– Error suppression refers to a greater response to the cascade context (a tone embedded in a predictable sequence) than to the deviant context (an unpredictable tone). We will be changing this term to “prediction signals”.

– Suppressed units (see Figure 3) refer to those units whose firing rate during tone presentation are lower than the preceding baseline period. We will be re-naming these units as “inhibited” units.

– Repetition suppression refers to a greater response to the standard context than the cascade context. Following the established convention in the literature, we will leave this term as is.

8. L. 155 "experimental" does this refer to recordings from IC? Also, please clarify if there were any differences in the results for SUA vs. MUA. This is particularly important for central IC, where single unit isolation is typically difficult.

We performed this analysis and believe that we can pool the units together in the manuscript. Figure 3—figure supplement 3 includes plots of the index of neuronal mismatch in laser off and on conditions for each of the subgroups in the central and shell regions of the IC separated by single (displayed in teal) and multi units (similar to Figure 3D,E,G,H, top panel and Figure 4C,E, top panel). No major differences exist in the distributions of these two groups, further justifying the decision to pool data from both for the analysis.

9. L. 707. "… stronger response … in a completely predictable sequence" this is an interesting point. If this is the case, should the response to a tone in a cascade differ from the many standards condition?

This is an interesting observation. The many standards condition, similar to the deviant condition, is completely unpredictable. However, this sequence does not contain the establishment or violation of a prediction, as is seen in the oddball sequence. It is possible that cells with negative prediction error are suppressed by prediction violation, which would explain their higher firing rates to the cascade than the deviant and equal responses to the cascade and many standards sequences.

All other reviewer suggestions have been addressed in the manuscript.

Reviewer #3:1. "additional response profiles were assigned for each tone in the oddball pair based on the number, timing, duration, and direction of peaks in the peristimulus time histogram. Neurons with only suppressive or offset responses were removed from the data set." These two sentences are unclear – what is a response profile? If this is a quantitative judgment, what were the criteria used? If qualitative, I'd suggest rephrasing to say that neurons were excluded by visual inspection of the PSTH to ensure that only neurons with excitatory peaks within a reasonable onset latency were included, and give parameters for what was considered a reasonable onset latency and duration.”

As mentioned in response #4 to reviewer #2, the categorization of sound responses (i.e., “response profiles”) was done quantitatively. We have edited lines 776-789 of the methods to include the following:

“Sound response profiles were categorized quantitatively from analysis of the combined responses to the standard and deviant tones using MATLAB’s “findpeaks” function with a minimum peak height set to the mean of the baseline period (50 ms before tone onset) +/- 3 standard deviations. Units that did not display maxima or minima during the tone duration period (0-50 ms) or in the 50 ms after (the “offset window”) were labeled as sound unresponsive and were removed from the analysis. Units that showed only a single minimum (“inhibited” units) or only a response in the offset window were similarly removed from the analysis. Units that showed at least one maxima during the tone duration period were included in the analysis and further categorized as either onset (single maxima in the first 10 ms after tone onset), sustained (single maximum after the first 10 ms after tone onset), E-I or I-E (units that displayed both a maximum and minimum during the tone duration period), biphasic (units that displayed two maxima during the tone duration period), or mixed (units with greater than 2 peaks during the tone response period). It was common for units to display a response both during the tone duration window and the offset window, and in these cases a combined response profile was assigned (e.g., onset/offset, sustained/inhibited-offset).

2. “Anesthetized vs. awake comparison – this is an important part of the study and I’d suggest mentioning it in the abstract (anything not mentioned in the abstract might as well not have happened….).”

We thank the reviewer for pointing out this omission and have edited lines 53-57 of the abstract to reflect the anesthetized vs. awake findings: “We also investigated how these metrics compare between the anesthetized and awake states by recording from the same neurons under both conditions. We found that metrics of predictive coding and deviance detection differ depending on the anesthetic state of the animal, with negative prediction error emerging in the central IC and repetition enhancement and prediction error being more prevalent in shell regions in the absence of anesthesia.”

3. “Central vs shell quantification by regression of best frequency vs depth – there is precedent for this method from the monkey and human literature; it might be worth citing this work to strengthen the justification for this choice.”

We thank the reviewer for bringing these important papers to our attention and have cited them in an updated version of the manuscript. We believe that this precedence in conjunction with our added histological data further strengthens the justification for using analytical methods to sort recording sites.

[Editors’ note: what follows is the authors’ response to the second round of review.]

The manuscript has been improved, and the reviewers agreed that this work is important, but there are some remaining issues involving the analyses that need to be addressed as outlined in greater detail below (Reviewers 2 and 3).Reviewer #2:This study measures the impact of feedback from auditory cortex (AC) to the auditory midbrain (inferior colliculus, IC) on neural encoding of predicable versus unpredictable sounds. Consistent with previous work, the authors find that inactivation of AC feedback reduces differential adaptation to high- versus low-probability stimuli. By introducing a new stimulus condition to their analysis, they provide evidence that the specific effect of AC feedback is to enhance prediction errors and that other forms of adaptation occur independent of AC.The authors have done a good job addressing concerns raised during the initial review. In particular they have more clearly contrasted their new results on prediction error with previous corticofugal/SSA work that did not distinguish between adaptation and prediction error. This provides a substantial advance on previous work.However, some concerns do persist around the validity of statistical methods used.L. 323-324: "To ensure that the laser-induced changes described above were opsin-mediated, we performed control experiments in two mice with identical manipulations to the experimental group, but in the absence of ArchT…"It feels like nagging, but the circumstances described in this study--where fairly symmetric tails of a distribution both shift toward zero--are exactly the case where regression to the mean is a concern. The control cited above provides evidence that activation of ArchT has some impact on adaptation, but it is not clear that this single control is adequate to support all the subsequent findings in the manuscript. It would be more convincing if the authors could categorize units based on responses averaged across laser-off and -on conditions. If this is not feasible, then the authors should address the following:

We thank the reviewer for this suggestion. We used well-established methods for estimating regression to the mean including using multiple baseline measurements and a control group, as we have done here. Notably, we do not observe any significant changes in the control group for any of the predictive coding metrics between laseroff and laser-on trials, suggesting that regression to the mean does not drive the effects in our experimental group. Categorizing the units on responses averaged across laser-off and laser-off conditions, as reviewer suggests, would not adequately parse out an effect due to regression to the mean from a true photo-suppression effect. In the instance that a real photo-suppression effect is present in the data, categorizing units in this way would lead to the biased selection of only those units that maintain similar responses to the standard and deviant on both laser-on and -off trials, i.e., those units that do not show photo-suppression effects.

1. Are the experimental groups in Figure 3S1B the same as in Figure 3B? Perhaps some details are missing or there is a labeling issue with the x axis? The distributions in 3S1B appear narrower than in 3B. A narrower distribution might indicate less noise, which would reduce the possibility of regression to the mean. Please clarify if in fact the experimental data should be the same and or if differences in the width of the iMM distributions might impact the validity of the control. Are the fractions of adapting/facilitating/non-adapting neurons similar for the control and experimental groups?

The data from the experimental groups in Figure 3S1B and Figure 3B are the same. These plots depict the SSA index for laser-off trials, computed from the mean firing rates over 45 trials in response to a tone in either the standard or the deviant context. Whereas indeed, a narrower distribution of trial-by-trial firing rates for an individual unit could signify less noise in the responses, therefore reducing the possibility of regression to the mean, we do not believe the same interpretation applies for these plots, as they do not depict measures across multiple trials (i.e., test/re-test data). A narrower distribution in this case would indicate that the population shows less extreme adaptation/facilitation and that units respond more similarly to tones regardless of statistical context. However, this is not what we observe here. Rather, we find that the width of the distributions is relatively matched between the control and experimental groups in Figure 3S1B, which suggests that the range of SSA indices in the control sample is similar to the experimental group, thus enhancing the validity of the control.

We also find similar proportions of adapting/facilitating/non-adapting neurons in the control groups (central: 23% adapting, 5% facilitating, 71% non-adapting; shell: 29% adapting, 18% facilitating, 53% non-adapting) compared to the experimental groups (see Figure 4A; central: 24% adapting, 6% facilitating, 70% non-adapting; shell: 29% adapting, 9% facilitating, 62% non-adapting), and have added these comparisons to the edited manuscript (lines 332-336).

2. While the control experiment/analysis supports conclusions reported in Figure 3, it is not clear that it is adequate to support the conclusions in the subsequent analyses, where the data are further processed (e.g., Figure 4C, IRS<0 only) or different quantities are analyzed (e.g., firing rate in Figure 5). One solution would be to run the same analysis on control data in each case. This option does seem cumbersome, and authors might have a better idea for how to address this concern.

The data were analyzed, as depicted in Figure 4C, in order to determine whether changes in repetition suppression (iRS > 0) or repetition enhancement (iRS < 0) drive the statistically significant change in the overall iRS metric in central non-adapting units (see Figure 4C, third row). We have run this comparison in the control data as well and did not observe a laser effect for the overall iRS metric (Figure 4C) , repetition suppression (p=0.16, paired t-test) or enhancement (p=0.099, paired t-test). This analysis has been added to Table 2 of the edited manuscript.

The firing rate analysis in Figure 5 was performed to further characterize whether the changes in predictive coding metrics with laser photo-suppression arise from changes to the standard, cascade, and/or deviant contexts. Given that no significant changes were seen in these metrics in the control group, we did not perform an additional characterization.

L. 88. "However, it remains unknown whether these modulations in the SSA index with cortical deactivation reflect changes in predictive processing." Minor. This sentence seems out of place, as the relationship between SSA and prediction error is not laid out until the next paragraph.

We have removed this sentence and further edited the Introduction section per the suggestions of Reviewer 3.

L. 95. "Prediction error…" It might help to rephrase this sentence to provide a definition of prediction error as a component of SSA, in the same way that the previous sentence defines repetition suppression.

We have edited this sentence to the following: “Repetition suppression is characterized by a decrease in firing rate to each subsequent presentation of a standard tone while prediction error signals an enhanced response to a deviant tone”.

L. 253 "… while an iPE value…" Should this be "… while the mean iPE value"? Also, since the same neurons were recorded in both conditions could a paired test be used here? Students T usually treats the two distributions as independent.

We thank the reviewer for this correction and have changed it to “mean iPE” in the text. Though the units were recorded in both the awake and anesthetized conditions, some units showed sound-evoked responses in only one of the two conditions that were either inhibited, offset, or non-responsive, and were subsequently removed from the analysis. Therefore, paired data for both conditions was not available for all units, and an unpaired test was used instead.

Reviewer #3:[…]Together, the results from this study will contribute to our understanding of predictive coding in the central auditory system. The strength of this study is the rigorous experiments and comprehensive analysis: the authors examined SSA in neurons from distinct IC subdivisions in both awake and anesthetized mice, assessed the specific effects of corticocollicular projections and analyzed previously-ignored facilitation. Addressing the following concerns will greatly increase the significance of the findings:1. The authors pool together data from single and multi-unit recordings. Although the inclusion of multi-units might not necessarily affect the described results, this limitation should be clearly stated in the "technical considerations" section of the discussion.

We thank the reviewer for this suggestion, and have added the following to lines 662-666 of the Discussion:

“The analyses in the present manuscript were performed on pooled single- and multi-unit data. Although we observed no differences in the iMM distribution between single- and multi-units (Figure 3 —figure supplement 3), the results of the present study should be interpreted with this limitation in mind. Namely, photosuppression-induced changes at the level of individual units may not reflect changes in single neurons.

For instance, the inclusion of multi-unit recordings may underlie the finding of "mixed" firing types in the IC, as shown in Figure 1-S2C.

The example of a “mixed” firing type shown in Figure 1-S2C is from a single unit. We have added the unit type (single or multi-unit) for each example depicted in Figure 1-S2 to the figure legend.

Importantly, the figure comparing responses between single- and multi-unit responses (provided in the response to the reviewers) should be included as a supplementary figure.

We thank the reviewer for this suggestion and have included the figure as a supplement (Figure 3—Figure Supplement 3).

Moreover, the author's interpretation of Figure 6 is that 'individual neurons exhibit distinct combinations of iPE and iRS'. Unless these data are only representing single units (which would not be consistent with the total number of single-unit recordings shown in the plot given to the reviewers), Figure 6 (panels A,B) fails to demonstrate that single neurons are exhibiting distinct iPE and iRS. This limitation should be clear in the Results section and included in the 'Technical Considerations' section of the Discussion.

We apologize for this error—this statement was meant to read “individual units exhibit distinct combinations of iPE and iRS”. The reviewer is correct that the data in Figure 6 are from both single and multi-units, and thus that we cannot conclude that individual neurons are exhibiting distinct iPE and iRS. We have edited this sentence and further discussed these limitations in the Discussion.

2. The data shown in Figure 1-S2 are not convincing that the recordings were obtained from either the central or shell IC. The authors should provide more histological evidence if the data are available. For example, the panel showing the recording site within the central IC (Figure 1-S2D) also shows some DiA signal in the shell. The authors claim to have performed histological reconstruction by DiD/A probe coating in a subset of animals (lines 708-709). The authors should state how many recording sites used in the study were evaluated histologically, and include all data in a supplemental figure.

Given that the shell IC surrounds the central IC dorsally, our electrode insertion sites (orthogonal to the dorsal surface of the IC) will always pass through the shell IC, even if the recording site is in the central IC. The source of dye in the shell IC in Figure 1-S2D is likely due to this initial insertion. The fact that histological assessment can render inconclusive subdivision assignments motivated us to pursue further categorization of the electrode sites based on the response properties of IC neurons, a method that has been established previously. We have added further details regarding the number of recording sites used for histological comparison in line 730.

In addition to the histology, the authors also use the response properties of the IC neurons to distinguish between central and shell recordings. This is a nice complementary method. However, while there are sites clearly distinguished as central or shell recordings based on sparseness and correlations between BF and depth, some sites are borderline (Figure 1-S2C). For example, several sites with high mean sparseness (characteristic of central IC) are categorized as shell recordings based on low correlations between BF vs depth. Could these sites instead be recordings in central IC with an electrode penetration angle slightly off the tonotopic axis? The differences between central and shell IC neurons in predictive coding, effects of anesthesia, and effects of cortico-collicular silencing are interesting findings. If the authors could provide additional data to give us more confidence in their ability to distinguish these sites, these results would be more compelling.

Given that our electrodes are inserted orthogonal to the dorsal surface of the IC and that the tonotopic gradient in the central nucleus runs dorso-lateral to ventro-medial, a slightly off angle penetration would likely still capture the increasing BF with depth relationship in this nucleus. It is not surprising to have some shell sites with higher sparseness values, as this metric can be affected by multiple tuning characteristics, such as inhibited regions of the tuning curve. Despite these few borderline cases, the BF vs. depth correlation and sparseness metrics show clear trends for parsing each IC sub-region.

3. Figure 1-S1C shows an off/rebound response for cortico-collicular neurons when the laser is turned off. The iMM (index of neuronal mismatch) is calculated by comparing the firing in response to a standard and a deviant tone that occurs 200ms after the standard. Thus, rebound activation of the cortico-collicular neurons after the last standard stimulus may alter the response to the subsequent deviant stimulus, impacting iMM. This potential confound should be discussed within the 'technical limitations' section of the Discussion.

In order to avoid potential laser rebound effects, the iMM was not computed by comparing immediately subsequent standard and deviant tones on laser trials. Rather, for a given last standard and subsequent deviant pair, only one of the tones was paired with the laser, and the pairing was switched on the next last standard/deviant presentation. This “block” approach to stimulus presentation and calculation of the iMM is described in lines 775782 of the text.

4. Figure 1-S1A. Although the study emphasizes the specific manipulation of the cortico-collicular projections, it should be noted that auditory corticofugal neurons that innervate the IC have other widespread targets including the medial geniculate body, striatum and lateral amygdala (Asokan et al., 2018, Nature Communications). Did the authors also see expression of axons in widespread brain regions with their viral strategy? If so, this could be shown in this panel or a supplemental figure. The involvement of these other regions in the inactivation studies should be addressed in the Discussion.

Consistent with Asokan et al. 2018, our viral strategy does label cortico-collicular collaterals in the medial geniculate body (see Blackwell et al. 2020) and additional downstream targets. The laser-induced changes seen in the present study are likely specific to the cortico-collicular pathway because they produce short-latency effects in the IC (Figure 3C, 3F), making muti-synaptic effects from other collateral sites unlikely. We have discussed this potential limitation in the Discussion in lines 667-674.